# BaCoN (Balanced Correlation Network) improves prediction of gene buffering

Thomas Rohde [1], Talip Yasir Demirtas[1], Sebastian Süsser[2], Angela Helen Shaw[1], Manuel Kaulich [2] & Maximilian Billmann [1]✉

## Abstract

Buffering between genes, where one gene can compensate for the loss of another gene, is fundamental for robust cellular functions. While experimentally testing all possible gene pairs is infeasible, gene buffering can be predicted genome-wide under the assumption that a gene's buffering capacity depends on its expression level and its absence primes a severe fitness phenotype of the buffered gene. We developed BaCoN (Balanced Correlation Network), a post hoc unsupervised correction method that amplifies specific signals in expression-vs-fitness correlation networks. We quantified 147 million potential buffering relationships by associating CRISPR-Cas9-screening fitness effects with transcriptomic data across 1019 Cancer Dependency Map (DepMap) cell lines. BaCoN outperformed state-of-the-art methods, including multiple linear regression based on our compiled gene buffering prediction metrics. Combining BaCoN with batch correction or Cholesky data whitening further boosts predictive performance. We characterized 808 high-confidence buffering predictions and found that in contrast to buffering gene pairs overall, buffering paralogs were on different chromosomes. BaCoN performance increases with more screens and genes considered, making it a valuable tool for gene buffering predictions from the growing DepMap.

**Keywords** Genetic Buffering; Prediction; Paralogs; Cancer Dependency Map
**Subject Category** Computational Biology

## Introduction

To ensure the robustness of cellular functions, organisms across all phyla rely on genetic redundancy where one gene can compensate for another gene's loss of function. This genetic redundancy is more complex in higher organisms and genome duplication throughout evolution coupled with the retention of genes if beneficial, gave rise to a large number of paralogs (Singh and Isambert, 2019). While the concept of genome duplication, retention of genes, and

subsequent sequence divergence is an attractive model to predict functional redundancy leading to functional buffering (Ryan et al, 2023b; De Kegel and Ryan, 2023), recent studies have shown that, while strongly enriched, the vast majority of paralogs do not functional buffer each other (Esmaeili Anvar et al, 2024). At the same time, functional buffering in a wider sense can also be observed between non-paralogs (Ryan et al, 2023a; Kryukov et al, 2016; Behan et al, 2019). For instance, a cell can cope with either loss of *BRCA1* or *PARP1*, but the combinatorial perturbation of the two genes is synthetic lethal (SL), resembling one of the most well-known clinical implications of functional buffering where *PARP1* inhibition can help patients harboring *BRCA1* mutations (Farmer et al, 2005). While isogenic CRISPR screens allow for testing the genome against one specific mutation, the all-by-all gene search space cannot be experimentally investigated (Oser et al, 2019; Williamson et al, 2016; Aregger et al, 2020; DeWeirdt et al, 2021).

Genetic networks from functional genomics data can assign function to genes in a systematic fashion. The similarity of expression signatures (co-expression) or genetic interaction signatures between pairs of genes has been exploited to classify genes by their function in model organisms (Costanzo et al, 2016; Fischer et al, 2015; Collins et al, 2007; Eisen et al, 1998; Billmann et al, 2018; Costanzo et al, 2010). Following this concept, co-essentiality networks in human cells derived from genome-scale CRISPR screens from the Cancer Dependency Map (DepMap) enable systematic prediction of gene function as well (Wainberg et al, 2021; Hassan et al, 2023; Pan et al, 2022). The DepMap perturbs all approximately 18,000 genes in the human genome, followed by the measurement of the effect of the perturbation on cell fitness. To date, the DepMap has screened more than 1000 cultured human cell lines and aims to expand this effort to several thousand cell models (Boehm et al, 2021). Experimental efforts are accompanied by a continued improvement of the analytical pipeline (Meyers et al, 2017; Dempster et al, 2021; Iorio et al, 2018; Vinceti et al, 2023), which adjusts the data for various factors that would otherwise compromise the interpretation of per-cell line fitness effects as well as co-essentiality networks that measure the similarity of fitness effects between genes across all cell lines. A prominent artifact is copy number-driven DNA-cutting toxicity effects that confound fitness scores and create a syntenic gene clustering bias (Wainberg et al, 2021; Meyers et al, 2017; Lazar et al, 2024; Vinceti et al, 2024). Those analytical pipelines have been

[1]Institute of Human Genetics, University of Bonn, School of Medicine and University Hospital Bonn, Bonn 53127, Germany. [2]Institute of Biochemistry II, Faculty of Medicine, Goethe University Frankfurt, Theodor-Stern-Kai 7, 60590 Frankfurt am Main, Germany. ✉E-mail: m.billmann@uni-bonn.de

instrumental for extracting gene fitness effects from every screened cell line.

A separate class of data normalization methods exploited the CRISPR screening-derived gene fitness effects to generate co-essentiality networks that predict the similarity of gene functions (Hassan et al, 2023; Wainberg et al, 2021; Rahman et al, 2021; Boyle et al, 2018). A comparison of different normalization strategies showed that the perhaps most crucial part of the normalization strategy is the whitening of the fitness effect gene-by-cell line matrix prior to the computation of between-gene association indices (Gheorghe and Hart, 2022).

More recently, to predict gene buffering, DepMap gene fitness effects have been contrasted with the mRNA expression in matched panels of cell lines. Conceptually, such a contrast would show positive association indices if the potentially buffered gene is more essential in cell lines in which the buffering gene is less expressed (De Kegel et al, 2021; Köferle et al, 2022; Pacini et al, 2024; Krieg et al, 2024; Lord et al, 2020) (Fig. 1A). In this case, gene fitness effects and mRNA expression must be measured in the same cell lines but in independent experiments. Multiple linear regression models are routinely deployed to estimate association indices while accounting for confounding factors such as cell lineage or growth media conditions (Pacini et al, 2024; De Kegel et al, 2021). Other powerful methods that perform data normalization prior to computing the association indices, such as data whitening, may carry the risk of unlinking biologically relevant covariation between the datasets. The benefit and risk of such a priori data normalization methods when predicting gene buffering from the DepMap have not been explored.

Here, to avoid the risk of unlinking covariation in orthogonal datasets, we developed BaCoN (Balanced Correlation Network), a method to correct correlation-based networks post hoc (Fig. 1A). BaCoN takes a correlation matrix and adjusts the correlation coefficient between each gene pair by balancing it relative to all coefficients each gene partner has with all other genes in the matrix. We assembled six sets of metrics to evaluate systematic buffering prediction performance. Based on those metrics, we benchmark BaCoN and compare it to multiple linear regression and simple correlation coefficients with and without a priori data normalization. We find that BaCoN alone outperforms the other methods, including a vastly increased performance over multiple linear regression. We identify a priori normalization methods suitable for buffering predictions from orthogonal datasets and demonstrate that adding the post hoc method BaCoN further improves performance, especially when predicting an extended set of buffering gene pairs. Finally, we generate a new, high-quality set of 808 gene pair buffering predictions.

## Results

### Genome-wide prediction of gene buffering

Genome-wide gene buffering can be predicted by defining cell lines where one gene's function is likely to be absent due to low expression or a loss-of-function mutation and then test if another gene displays more severe fitness effects in just those cell lines (Köferle et al, 2022; De Kegel et al, 2021; Krieg et al, 2024). We systematically evaluate state-of-the-art data normalization

methods, some of which have previously been shown to substantially improve co-essentiality networks (Hassan et al, 2023; Gheorghe and Hart, 2022; Wainberg et al, 2021), for predicting gene buffering from the Cancer Dependency Map (DepMap) and present a novel method to further improve such predictions. To build our method, we hypothesized that gene buffering predictions, in contrast to co-essentiality or co-expression-derived functional similarity predictions, have two fundamental characteristics. First, genes are likely only buffered by one or a few genes, often paralogs. Second, buffering predictions align two independent data matrices: gene perturbation fitness effects and mRNA expression, and independent a priori normalization of those input data potentially unlinks biologically meaningful covariation (Fig. 1A,B). In support of the second hypothesis, we found that different a priori normalization approaches of the input data weakened or completely removed the correlation of the most strongly correlated, experimentally validated buffering paralog gene pairs including *FAM50A-FAM50B*, *RPP25L-RPP25*, *EIF1AX-EIF1AY* and *DDX3X-DDX3Y* (Köferle et al, 2022; Thompson et al, 2021), which was not the case for co-essentiality networks derived from the same matrix of Chronos scores (Appendix Fig. S1A–G,J–L). Exploiting the few-buffering-gene hypothesis and avoiding the risk of unlinking covariation in independent input data, we developed BaCoN (Balanced Correlation Network), a method to correct correlation-based networks post hoc (Fig. 1A; Appendix Fig. S2A–D). BaCoN emphasizes specific high pairwise coefficients by penalizing values for pairs where one or both partners have many similarly high values (see "Methods" for details).

To evaluate the gene buffering prediction performance of BaCoN and alternative normalization strategies, we utilized DepMap 23Q2 fitness scores and the associated CCLE expression data (Tsherniak et al, 2017; Meyers et al, 2017; Barretina et al, 2012). DepMap fitness (Chronos) scores measure how much the fitness of each of the 1095 tested cell lines declines upon CRISPR-Cas9-mediated perturbation of each of the roughly 18,000 protein-coding genes. The CCLE expression data contains log2 TPM + 1-transformed mRNA expression data for 19,193 genes in each of the 1019 cell lines that intersect with the 23Q2 DepMap gene effect data. We restricted our analyses to genes that were expressed at an average log2 TPM + 1 > 1 across all cell lines. Overall, this left 12,401 potentially buffering genes (CCLE expression data matrix) and 11,885 potentially buffered genes (DepMap Chronos matrix). Notably, in contrast to co-expression or co-essentiality analysis, (i) the potentially buffered and buffering genes do not have to be identical and (ii) the correlation between gene A and gene B does not equal the correlation between gene B and gene A (Fig. 1A). All a priori normalizations, such as batch correction, were performed on these two matrices while the post hoc method BaCoN was computed on the PCC network derived from those matrices (unless stated otherwise).

### Metrics for evaluation of buffering prediction from the Cancer Dependency Map

To evaluate the performance of BaCoN and various state-of-the-art methods for their ability to predict gene buffering, we defined six metrics. First, since systematic, genome-wide searches for buffering gene pairs are expected to be enriched for paralog pairs (De Kegel

et al, 2021; Köferle et al, 2022), we defined the number of Ensembl paralog gene pairs with at least 20% sequence identity as well as the number of Ohnologs among the top buffering predictions as the first metric (Appendix Fig. S2E) (Singh and Isambert, 2019; Yates et al, 2019). Second, we counted the number of predicted, predicted, and validated (Appendix Fig. S2F) as well as experimentally identified (Appendix Fig. S2G) synthetic lethal paralog pairs from previous studies (De Kegel et al, 2021; Ito et al, 2021; Esmaeili Anvar et al, 2024; Thompson et al, 2021; Dede et al, 2020; Gonatopoulos-Pournatzis et al, 2020; Parrish et al, 2021). Based on

those first two sets of metrics, we defined the top 100 and top 1000 gene pairs as the strict and standard buffering predictions (Appendix Fig. S2H,I) (see "Methods" for details). Third, we considered the ability to detect a gene effect's dependency (or addiction) on its own expression level, which we refer to as self-addiction (Pacini et al, 2024). Fourth, as mentioned above, we reasoned that genes are buffered by one or few other genes and that the prediction of a large number of buffering genes may be due to confounding factors. In support of this hypothesis, we found that an uncorrected correlation of expression and fitness scores

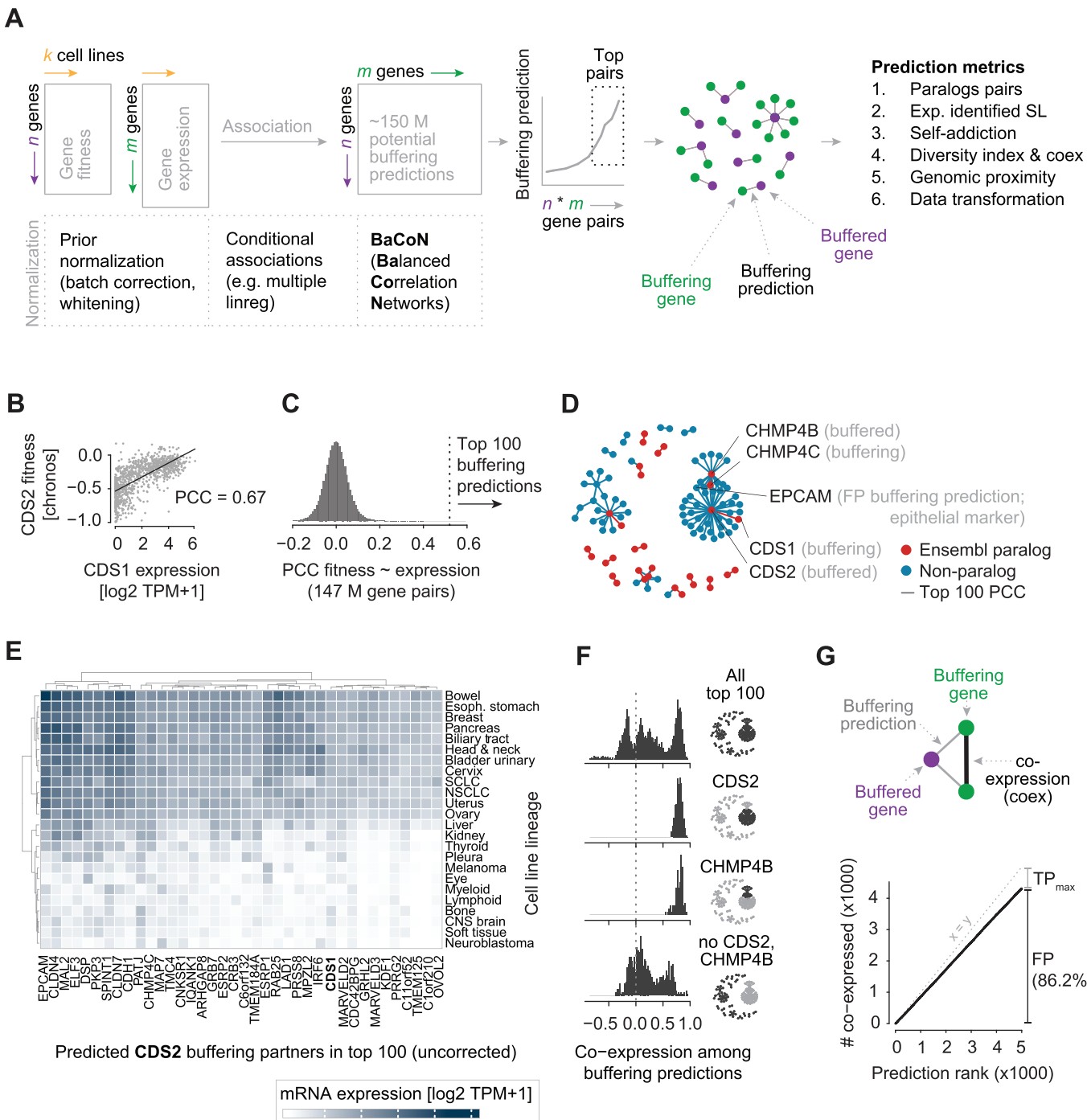

◀ **Figure 1.  Normalization strategies for predicting gene buffering from the Cancer Dependency Map.**

(**A**) Schematic illustration of normalization strategies where gene buffering is predicted by associating each n gene fitness effect from CRISPR screens with each m gene mRNA expression in k cell lines. N and m can be different but k needs to be matched in both datasets. (**B**) Correlation of *CDS1* expression with *CDS2* perturbation fitness effects (Chronos scores) in 1019 23Q2 cell lines. PCC Pearson's correlation coefficient. (**C**) Distribution of all pairwise ~147 million PCCs between 12,401 potentially buffering genes (CCLE expression data matrix) and 11,885 potentially buffered genes (DepMap Chronos matrix). (**D**) Network of the PCC-based top 100 buffering predictions. The epithelial marker *EPCAM* is a likely false positive (FP) buffering prediction due to its tissue-type-driven co-expression with the buffering paralog partners *CDS1* or *CHMP4C*. (**E**) Mean expression in cell lines derived from the indicated lineages for all genes connected to *CDS2* among the top 100 buffering predictions shown in (**D**). (**F**) Co-expression of the buffering genes of the top 100 predictions with and without the genes connected to *CDS2* and/or *CHMP4B*. Co-expression is measured as the PCC of log2 TPM + 1 values in 1019 cell lines. The network thumbnail (**D**) highlights the buffering genes included in a given histogram. (**G**) Number of likely FP buffering predictions due to co-expression with a buffering gene. Shown are the top 5000 PCC-based predictions. A co-expressed FP was identified as a gene that (i) buffers a gene that is more strongly buffered by a second gene, which (ii) is co-expressed with the FP gene (see "Methods" for details).

predicted some genes to be buffered by a large number of genes among the top 100 buffering predictions of ~147 million tested pairs (Fig. 1C,D). The fact that (i) among those buffering genes was a paralog partner of the buffered gene, (ii) this paralog partner showed strong tissue-specific expression, and (iii) the buffering genes were strongly co-expressed suggested non-buffering false positive prediction of buffering genes when considering correlation without correction. For instance, cell line dependency on *CDS2* showed a strong correlation with *CDS1* expression. However, *CDS1* was co-expressed with the epithelial marker gene *EPCAM* and several other genes with epithelial expression patterns (Fig. 1E,F). The same was true for *CHMP4B*, which showed a high dependency on the expression of *CHMP4C*, another gene with higher expression in cell lines of epithelial origin (Fig. 1F; Appendix Fig. S3A). We concluded that those scenarios show high correlation between the gene effect of one gene with the expression of many without a direct buffering relation. To quantify such a bias, we reported the minimal number of genes accounting for 50% of predictions as a diversity index between 0 and 1, where 1 indicates that no gene occurs among the predicted pairs more than once (very diverse) (Appendix Fig. S3B–D). While this diversity index provides a simple metric to judge the relative impact of co-expression-driven likely false positive (FP) buffering predictions, we also counted the number of gene pairs. Genes whose expression was predicted to buffer a perturbed gene were considered FP when they showed co-expression with a gene predicted to buffer the same gene but with higher confidence (see methods for details), accounting for 86.2% of the top 5000 buffering predictions when using the PCC to compare expression and gene effect data (Fig. 1G). Fifth, the effect of genomic location on CRISPR screens can be substantial (Meyers et al, 2017; Vinceti et al, 2024; Lazar et al, 2024). While the DepMap 23Q2 gene effect data has been corrected for what we refer to as proximity-driven effects and PCC-based gene buffering predictions show that this removed the increased numbers detected when using uncorrected data, several of the normalization methods we test in this work showed a strong proximity bias (Appendix Fig. S3E–G). Based on this observation, we defined gene pairs encoded on the genome within 10 Mbp of each other as potential FP predictions. Lastly, we assessed how strong a normalization method transformed the original correlation data, with the goal of changing the data as little as possible while removing biases. Specifically, we tested the proportions of pairs within the top buffering predictions after normalization that lacked a strong correlation in the original data. Since, before normalization, 788,051 gene pairs showed a *z*-score-transformed correlation larger than 3 (density of 0.005%), we defined this pool of gene

pairs as the preferred set for the top 100 predictions after normalization. Gene pairs with a *z*-score-transformed correlation smaller 0, 1, 2, or 3 were all defined as uncertain without in-depth exploration (Appendix Fig. S1G–N).

To streamline the interpretation of those six metrics, we defined gene pairs driving those metrics as true positive (TP) or false positive (FP) buffering predictions whenever plausible. We labeled predicted gene pairs in metric groups one to three as standard TP and groups one and two as stringent TP. FP predictions were derived from metric groups four and five, and the FDR was computed as FDR = TP/(TP + FP).

## BaCoN outperforms existing methods to predict gene buffering

BaCoN is a post hoc normalization method for predicting gene buffering by comparing the perturbation effects of a gene and mRNA expression of another gene. In its standard implementation, a Pearson's correlation coefficient (PCC) matrix between gene effects and mRNA expression is computed first. Overall, applying BaCoN on this PCC matrix (network) improved the first five metrics, while not negatively affecting the sixth (data transformation strength). Among the top 100 buffering predictions, BaCoN increased the number of Ensembl and Ohnolog paralog pairs from 16 to 71 and 10 to 53, respectively (Fig. 2A; Dataset EV1). Predicted synthetic lethal (SL) paralog pairs by De Kegel and colleagues increased from 8 to 36 and experimentally identified SL paralog pairs from 7 to 34, 3 to 21, 0 to 5 and 3 to 10, respectively (Fig. 2A). BaCoN substantially reduced the bias as shown by an increased diversity index, reduced co-expression of buffering genes and increased identification of self-addictions (Fig. 2B,C,E). For instance, while *CDS2* and *CHMP4B* were represented in 38 and 15 of the top 100 PCC-based predictions, respectively, the only top 100 pair for *CDS2* was with its paralog partner *CDS1* and *CHMP4B* dropped out of the top 100, with the *CHMP4C* being the strongest partner after BaCoN normalization. At the same time, BaCoN only moderately transformed the PCCs, only elevating gene pairs into the top 100 predictions that had a PCC *z*-score larger than 3 but also an elevated number of buffering predictions between genes with genomic proximity (Fig. 2D,F; Appendix Fig. S1H). Together, this showed that balancing the PCC-based network substantially improved the performance to predict gene buffering.

To compare BaCoN performance at removing spurious, likely cell line lineage-driven buffering predictions as described for *CDS2* and *CHMP4B* (Fig. 1D–G; Appendix Fig. S3A), we tested several established supervised correction methods. We performed batch

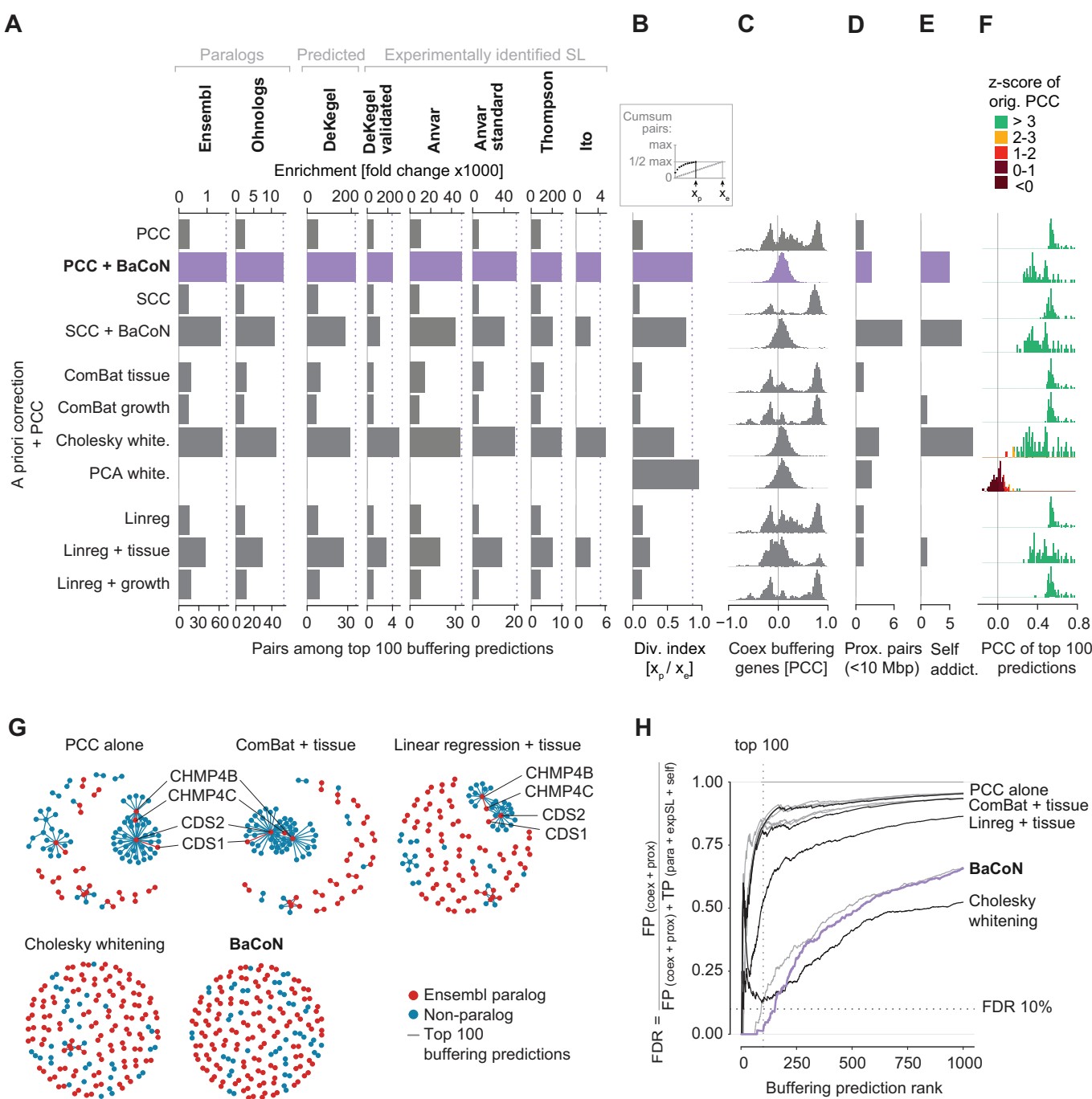

**Figure 2.  BaCoN outperforms existing methods for predicting gene buffering.**

(A) Performance of BaCoN and alternative normalization methods to systematically predict functional buffering. The performance is evaluated using the enrichment (fold change; fc) and absolute number of sequence-based paralogs, predicted synthetic lethalities (SL), and experimentally identified SLs between paralogs among the top 100 buffering gene pair predictions of each method. Approximately 147 million gene pairs are tested across 1019 different cell lines from the DepMap. PCC Pearson's correlation coefficient, SCC Spearman's correlation coefficient, Linreg multiple linear regression with indicated experimental factor as a covariate. (B) Diversity index among the top 100 buffering predictions. Co-expression is measured across the top 100 buffering predictions. (C) Co-expression among all buffering genes included in the top 100 buffering predictions. Co-expression is measured across the 1019 cell lines used for the buffering predictions. (D) Number of buffering gene pairs that are encoded in genomic proximity (<10 Mbp). (E) The number of identified self-addictions where a gene's fitness effect is predicted to depend on its expression. (F) The strength of the data transformation of each method is indicated by the number of pairs with low z-score-transformed PCCs. This PCC describes the gene fitness effect-vs-expression association and is identical to the PCC shown in (A). (G) PCC-based network of the top 100 buffering predictions of five selected methods. (H). False discovery rate (FDR) of the top 1000 predictions made by each method shown in (A–F). The methods shown in (G) are colored black and violet. For computing the FDR, true positive (TP) predictions were defined as predicted buffering between paralogs (para), experimentally discovered synthetic lethal pairs (expSL) as well as self-addictions (self). False positive (FP) predictions were defined as predicted buffering between proximal (prox; <10 Mbp) gene pairs and those driven by co-expression (coex) between buffering genes.

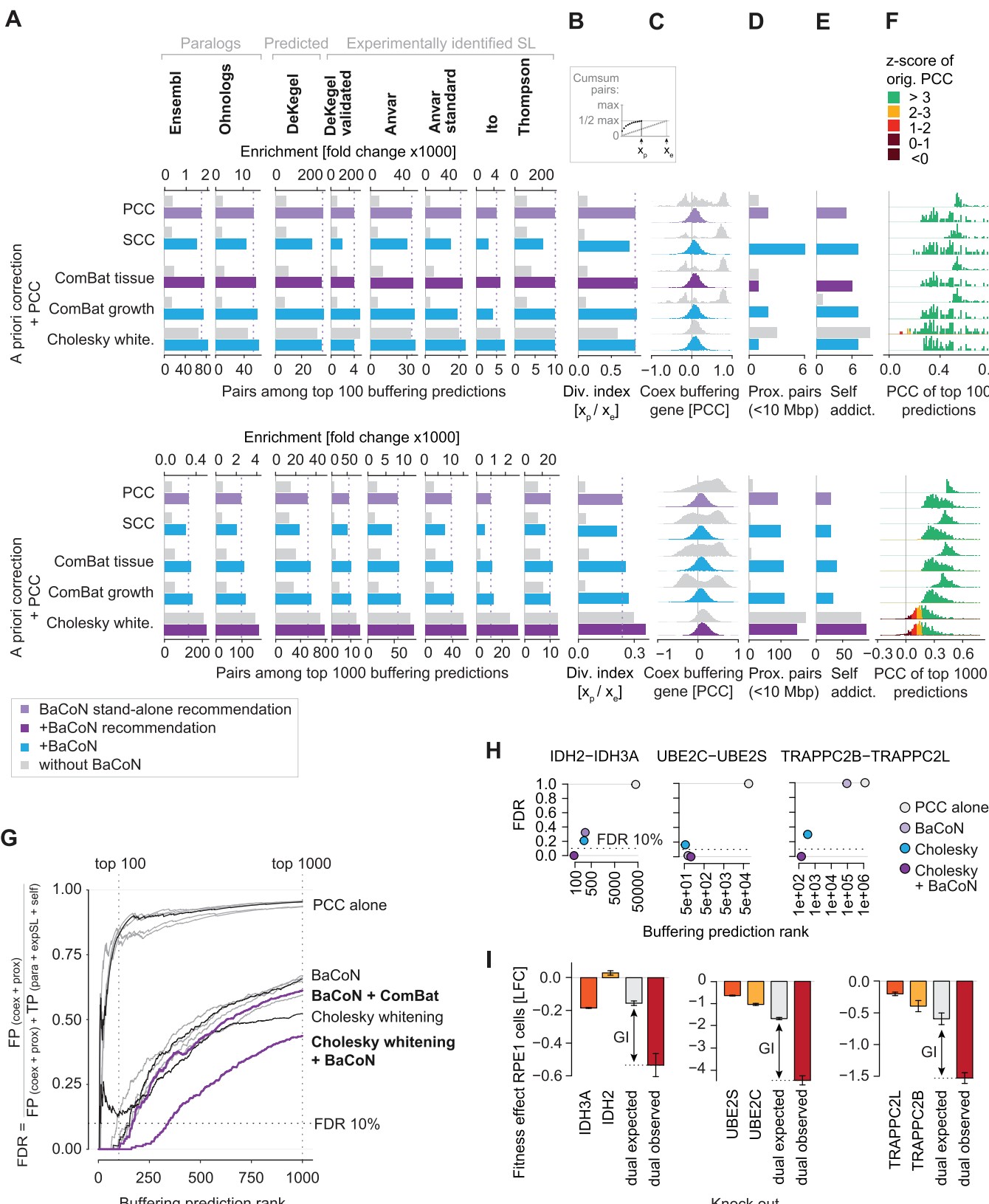

**Figure 3.  Adding BaCoN improves the prediction of gene buffering.**

(A) Performance of BaCoN in combination with alternative normalization methods to systematically predict functional buffering. The performance is evaluated using the enrichment (fold change) and absolute number of sequence-based paralogs, predicted synthetic lethalities (SL), and experimentally identified SLs between paralogs among the top 100 and top 1000 buffering gene pair predictions of each method. Approximately 147 million gene pairs are tested across 1019 different cell lines from the DepMap. PCC Pearson's correlation coefficient, SCC Spearman's correlation coefficient, Linreg multiple linear regression with indicated experimental factor as covariate. (B) Diversity index among the top 100 buffering predictions. (C) Co-expression among all buffering genes included in the top 100 buffering predictions. Co-expression is measured across the 1019 cell lines used for the buffering predictions. (D) Number of buffering gene pairs that are encoded in genomic proximity (<10 Mbp). (E) The number of identified self-addictions where a gene's fitness effect is predicted to depend on its expression. (F) The strength of data transformation of each method is indicated by the number of pairs with low z-score-transformed PCCs. This PCC describes the gene fitness effect-vs-expression association and is identical to the PCC shown in (A). (G) False discovery rate (FDR) of the top 1000 predictions made by each method. For computing the FDR, true positive (TP) predictions were defined as predicted buffering between paralogs (para), experimentally discovered synthetic lethal pairs (expSL) as well as self-addictions (self). False positive (FP) predictions were defined as predicted buffering between proximal (prox; <10 Mbp) gene pairs and those driven by co-expression (coex) between buffering genes. (H) Prediction rank and FDR of three selected gene pairs for PCC, BaCoN, Cholesky, and Cholesky plus BaCoN-based buffering predictions. (I) Experimental validation of the buffering capacity of the three selected gene pairs (H) in hTERT-RPE1 cells upon individual and dual CRISPR-Cas9 knockout (KO). The negative genetic interaction (GI), where the observed dual KO effect is more severe than expected from both individual KO, indicates a buffering effect between the tested gene pairs. hTERT-RPE1 cell fitness was quantified in a GFP competition assay by FACS after transduction with gRNAs targeting the respective genes and after 14 days. Log2 fold change (LFC) represents the change in cell numbers. Bars represent the mean LFC ($n = 3$), error bars the SEM.

correction using ComBat with cell line lineage as batch of the fitness effect and expression matrices separately followed by computing a PCC matrix on the corrected matrices (Dataset EV1). Counterintuitively, this increased the number of pairs including *CDS2* or *CHMP4B* to 39 and 31, respectively (Fig. 2G). In contrast, as expected multiple linear regression between gene effect and mRNA expression where lineage is the covariate reduced the number of pairs including *CDS2* or *CHMP4B* to 15 and 1, respectively (Fig. 2G). In conclusion, BaCoN reduced the bias of *CDS2* and *CHMP4B*, likely caused by lineage-driven co-expression of their paralog partner *CDS1* and *CHMP4C*, more effectively than both established methods that correct for lineage in a supervised fashion.

Next, we expanded the comparison of BaCoN to other methods including classical a priori supervised batch correction and noise whitening methods as well as multiple linear regression. We expanded the above-described supervised methods to correct for growth type, a feature distinguishing adherent, suspension, or mixed growth of each cell line. However, improvements regarding all metrics were less pronounced than for the lineage-based corrections (Fig. 2A–F). Finally, we turned to a method that we and others had independently confirmed to perform very well on co-essentiality networks—data whitening with subsequent computation of pairwise PCCs (Hassan et al, 2023; Gheorghe and Hart, 2022; Rahman et al, 2021). Data whitening can be done with different sphering methods. We were surprised to see that Cholesky whitening of the gene effect and mRNA expression matrices prior to computing the association indices performed almost as good as BaCoN regarding all metrics (Fig. 2A–G). However, Cholesky whitening strongly transforms the data, which led to several top predictions with low z-score-transformed PCCs, an issue that became dominant when considering larger numbers of buffering predictions (Fig. 2F; Appendix Fig. S4A,B). Moreover, Cholesky whitening predicted many genomically proximal gene pairs as buffering pairs, a bias Chronos fitness scores had been normalized for (Fig. 2D: Appendix Fig. S3E,F) (Vinceti et al, 2024).

To summarize the separately illustrated metrics and compare all methods, we computed the FDR with likely TP and likely FP buffering predictions as defined above (see "Methods" for details). This showed that BaCoN outperformed all other methods by predicting gene buffering, especially for the most highly ranked

predictions (Fig. 2H; Appendix Fig. S5A,B). Notably, Cholesky whitening started to slightly outperform BaCoN when predictions at lenient FDR thresholds were considered (Fig. 2H; Appendix Fig. S5A,B).

Together, BaCoN outperformed supervised correction methods by providing a favorable combination of high paralog pair and known SL predictive performance while only moderately transforming the data.

## Adding BaCoN improves the performance of priori normalization methods

BaCoN is a post hoc correction method that accounts for biases after association indices such as PCC-based gene buffering predictions have been computed. Since the state-of-the-art alternative methods perform data correction prior to or during association index computation (Fig. 1A), we tested if BaCoN could further improve the performance of already normalized scores. Adding BaCoN improved the performance of all methods and across all metrics (Fig. 3A–G). The exception is multiple linear regression, the most routinely deployed method for associating gene effect with expression data, since p-values are used instead of association indices (De Kegel et al, 2021; Pacini et al, 2024). While BaCoN alone already outperformed multiple linear regression (Fig. 2A–E), we did not fine-tune BaCoN at this point to consistently improve multiple linear regression association scores (p-values). A priori normalization methods such as batch correction or noise whitening showed increased performance even outperforming PCC plus BaCoN (Fig. 3A–D,E–G; Appendix Fig. S5C,D). For instance, normalizing gene effect and expression data using ComBat with defining batches by cell line lineage consistently predicted more buffering paralog pairs and experimentally identified SL paralog genes pairs than BaCoN alone and performed best overall when focusing on the strongest predictions (Fig. 3A,G). In contrast to Cholesky whitening, ComBat batch correction only made predictions for pairs with an original PCC z-score greater 3 (Fig. 3F; Appendix Fig. S1J,L; Dataset EV1). This suggested that a moderate batch correction prior to BaCoN normalization provides well-adjusted gene buffering predictions without a risk to mistakenly identify gene pairs lacking clear covariance in the original gene effect and expression data.

We initially hypothesized that independent normalization of the input data prior to computing association indices could unlink biologically relevant information between the datasets. While PCA-based whitening completely unlinked gene fitness effect and expression data (Fig. 2A–E,H; Appendix Fig. S1A,B,E,K), Cholesky whitening showed a surprisingly high number of overall and SL paralog pairs, and Cholesky whitening performance further increased when adding BaCoN (Fig. 3A; Dataset EV1). Notably, the more relaxed list of top 1000 predicted buffering pairs (Appendix Fig. S2H,I) showed even stronger performance of Cholesky whitening with BaCoN outperforming every other method in every paralog pair, diversity index and the number of self-addictions (Fig. 3A,B,E). In addition, adding BaCoN after Cholesky whitening reduced the otherwise high number of likely false positive buffering predictions with close genomic proximity (Fig. 3D), leading to an overall much lower empirical FDR compared to Cholesky whitening alone (Fig. 3G; Appendix Fig. S5C,D). This suggested that Cholesky whitening with BaCoN can help predict potential buffering pairs with high sensitivity while also promoting the detection of likely FP predictions.

Across all metrics, Cholesky whitening plus BaCoN substantially outperformed all other methods alone or in combination (Fig. 3A–G). To demonstrate how both Cholesky whitening and BaCoN impact the prediction of true buffering between pairs with (i) low correlation prior to normalization, (ii) no, to the best of our knowledge, described synthetic lethality, and (iii) lower empirical FDR in the combinatorial normalization as compared to Cholesky alone or BaCoN alone normalization, we selected three gene pairs for experimental validation in a GFP competition assay (Fig. 3H; Dataset EV1). Upon CRISPR-Cas9-based perturbation of the genes individually and in combination in cultured hTERT-RPE1 cells, we observed that dual perturbation of *IDH3A* and *IDH2*, *UBE2C* and *UBE2S*, and *TRAPPC2B* and *TRAPPC2L* led to a more severe fitness effect than expected (Fig. 3I). To further test how BaCoN improves buffering predictions when added after Cholesky whitening, we compared the scores for recently described synthetic lethalities. We found that the cancer patient treatment-relevant relation between *PELO* and *FOCAD*, or the experimentally identified interactions between *SLC7A2* and *SLC7A1*, or *SLC7A6* and *SLC7A1* were best predicted upon combinatorial normalization (Appendix Fig. S5E; Dataset EV1) (Borck et al, 2025; Wolf et al, 2024).

Together, adding BaCoN to established a priori normalization methods can further improve gene buffering predictions. We propose to combine BaCoN with ComBat batch correction for low risk, slightly improved predictive power and BaCoN with Cholesky whitening for higher risk, highly powered predictions of gene buffering.

## BaCoN gains power with growing datasets

The Cancer Dependency Map (DepMap) that we here used for predicting gene buffering has been an ongoing effort mapping gene dependency across various cancer models. More than 1000 different cell lines (models) have been screened, and additional screens are performed and released in regular intervals (Boehm et al, 2021; Arafeh et al, 2025). To anticipate the potential impact of BaCoN on the future DepMap, we compared its performance with different numbers of screens. We sub-sampled 100, 200, 500 screens

each 10 times from the 23Q2 DepMap set of 1019 screens and tested the performance of BaCoN and the different a priori normalization methods with and without BaCoN. When using an already substantial set of 100 cell lines, neither method outperformed the simple PCC (Fig. 4A–C; Appendix Figs. S6A–D and S7A,B). At 200 cell lines, all more sophisticated methods started to outperform the simple PCC, with BaCoN ComBat plus BaCoN, Cholesky whitening and Cholesky whitening plus BaCoN outperforming the other methods lines (Fig. 4A–C; Appendix Figs. S6A,C and S7A). The addition of BaCoN to Cholesky-whitened data became more beneficial at 500 cell lines, particularly with regard to preventing a high number of genomically proximal, likely FP predictions (Fig. 4A–C; Appendix Fig. S6A–D). In contrast to multiple linear regression, BaCoN alone or in combination with a priori normalization was able to increase the number of predicted paralog pairs between 500 cell lines and the full data (Fig. 4A–C). Since performance gains did not increase linearly with the number of screens, predictive power to find gene buffering may have started to converge at 1019 screens to a number we might see when testing a future dataset.

We reasoned that future screens added to the DepMap will likely provide cases of a buffering gene losing expression while being widely expressed in the current set of cell lines or the buffered gene displaying currently unseen strong fitness dependency. We tested the impact of such scenarios by testing different numbers of potential buffering and buffered genes by varying the minimal required number of cell lines with fitness effects and loss-of-expression. This spanned input data from 3.4 to 272.3 million gene pairs, including 1350 to 107,009 Ensembl paralog and 44 to 11,015 Ohnolog pairs (Fig. 4D,E). Among the tested gene pairs, Ensembl paralog or Ohnolog pairs showed higher density when more gene pairs were considered whereas predicted and experimentally identified paralog SLs showed lower density in that area, underlining the investigation bias in current studies (Fig. 4F). In those search spaces, similar to multiple linear regression and simple PCC, BaCoN correctly predicted more absolute and relative (fold change) SL pairs with increasing numbers of potentially buffering genes (Fig. 4G,H). When adding more potentially buffered genes, BaCoN tended to benefit from this increased search space more than the other methods. Together, BaCoN particularly gains predictive power in larger datasets, making it a useful tool for future studies in the growing DepMap.

BaCoN has been designed to improve gene buffering predictions through contrasting CRISPR-Cas9 gene perturbation effect with mRNA expression and particularly excels when this is done across a large number of cell lines. We tested other omics data from the DepMap portal including shRNA perturbation screening and protein expression data to detect potential buffered and buffering genes, respectively. While CRISPR-Cas9 gene effect and mRNA expression data showed by far the best performance, adding BaCoN was beneficial when contrasting other omics data to predict gene buffering (Appendix Fig. S8A). Most of the datasets provide additional information about protein-coding genes. We also tested if we could expand the list of potentially buffering genetic elements to non-coding (nc)RNAs. At a threshold determined for protein-coding pairs below (see "Methods" for details), we were able to add 52 gene-ncRNA buffering predictions, including the *MYC-MYC-NOS* pair, to the list of 808 high-confidence predictions (Appendix Fig. S8B–D; Dataset EV3).

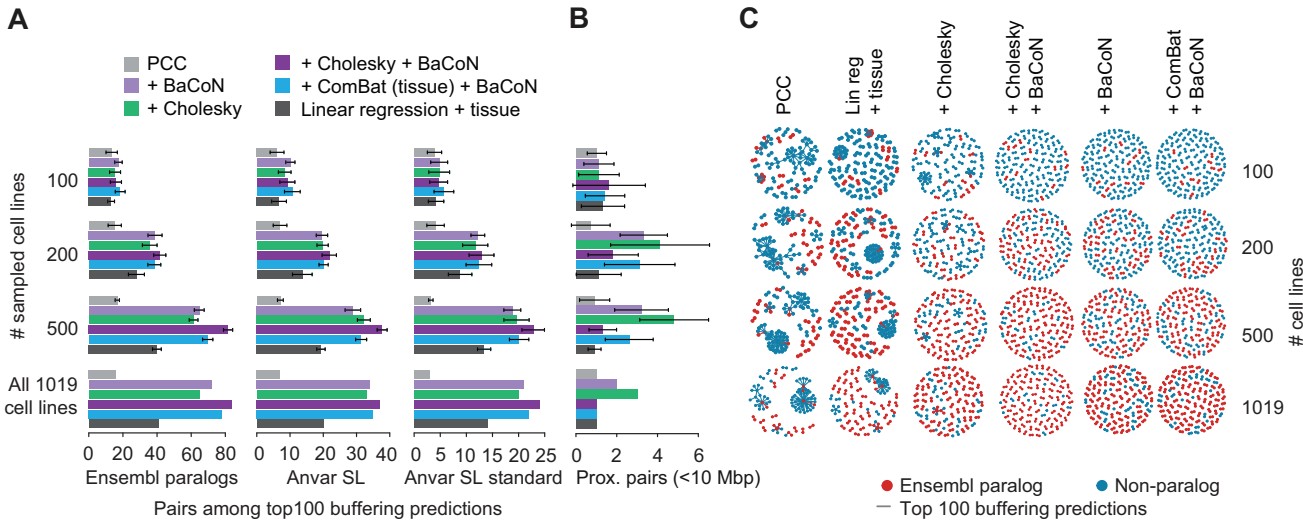

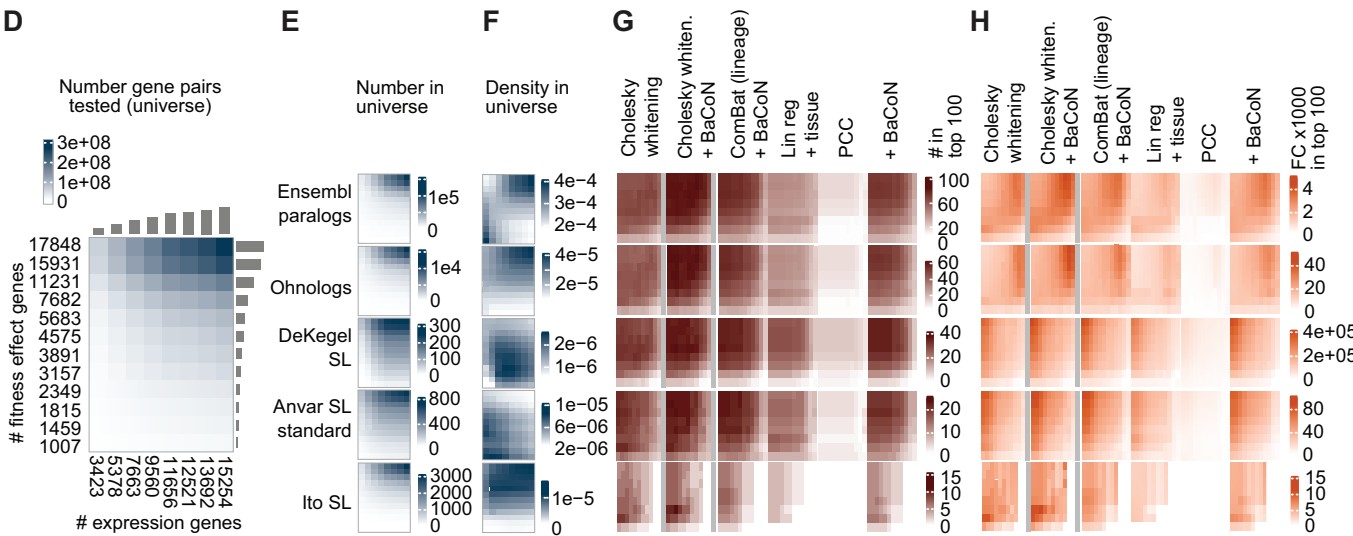

**Figure 4. BaCoN improves buffering predictions with a growing number of cell lines and genes tested.**

(A) Performance of BaCoN and selected alternative normalization methods to predict functional buffering. Performance is quantified as the absolute number of Ensembl paralogs and experimentally identified SL paralogs among the top 100 buffering gene pair predictions of each method. Approximately 147 million gene pairs are tested across 100, 200, 500 or 1019 different cell lines from the DepMap. 100, 200, and 500 cell lines were sampled from the 1019 cell lines 10 times, and bars represent the mean, and error bars represent the standard deviation of the counts. (B) A number of buffering gene pairs that are encoded in genomic proximity (<10 Mbp) in the data are also shown in (A). Error bars represent the standard deviation of the counts. (C) PCC-based network of the top 100 buffering predictions of five selected methods. (D) Gene pair search space when accepting different numbers of genes based on fitness effect and expression thresholds. (E) Numbers of paralog pairs in the different search spaces. (F) The density of paralog pairs in the different search spaces. (G) Numbers of different sets of paralog pairs among the top 100 buffering predictions via BaCoN, multiple linear regression or PCC in the different search spaces. (H) Enrichment (fold change; fc) of different sets of paralog pairs among the top 100 buffering predictions via BaCoN, multiple linear regression or PCC in the different search spaces.

## Characteristics of buffering gene pairs in the human genome

We present BaCoN as a complementary normalization method for predicting gene buffering from the DepMap, and suggest combining BaCoN with Cholesky whitening for sensitive predictions. Deploying this combination of methods, we assembled a list of buffering gene pairs at an empirical FDR of 50% with subsequent removal of gene pairs with genomic proximity or driven by co-expression between buffering genes (both FP), as well as self-

addictions (TP) (see methods for details). The remaining 808 high-confidence predictions contained 227 paralog pairs, many of which had previously been experimentally identified as well as 581 gene pairs without previous description of buffering (Dataset EV2). We first assessed how the 808 gene buffering predictions were covered by orthogonal datasets and analyses. In contrast to simple standard analysis methods of the same DepMap data, we predicted *YAP1* buffering its paralog *WWTR1* and *WWTR1* buffering *YAP1*, or *ARID1A* buffering its paralog *ARID1B* and vice versa, none of which showed the correlation between gene fitness effect and

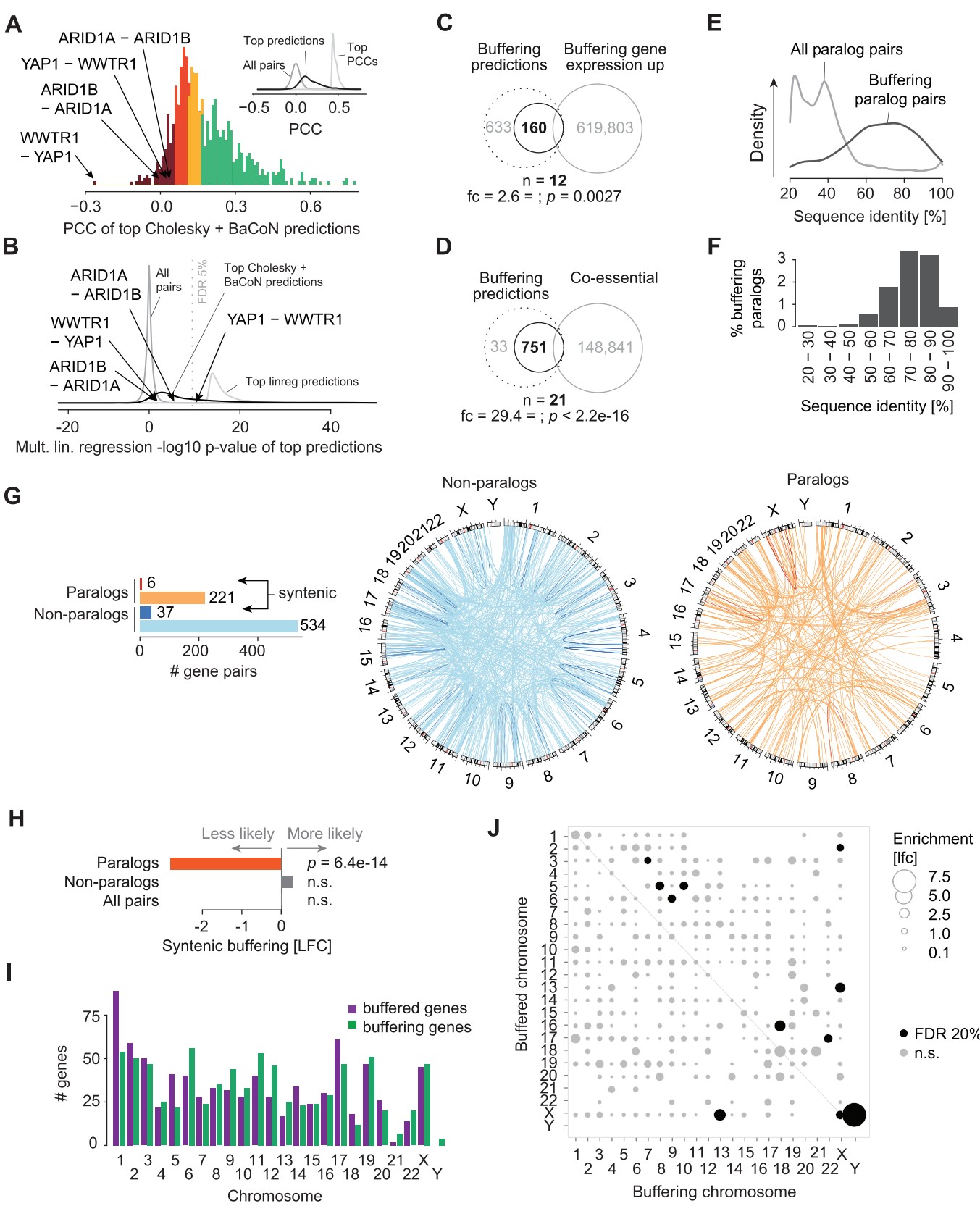

**Figure 5. Characteristics of buffering gene pairs in the human genome.**

In total, 808 high-confidence buffering gene pairs were predicted by combining Cholesky whitening with BaCoN. (**A**) Histogram of fitness effect-vs-expression PCCs of the 808 buffering gene pairs. Bars are colored by the PCC z-score bins ( > 0, 0–1, 1–2, 2–3, >3). Two paralog pairs, both the AB and BA pair, displayed low PCCs and were predicted among the 808 pairs are highlighted. The inset contrasts all 147 million PCCs (gray) with the top 1000 PCCs (light gray) and the PCCs of the 808 buffering pairs (black). (**B**) Multiple linear regression-based buffering prediction scores ($-\log_{10}$ p-value) of all 147 million gene pairs (gray), the top 1000 pairs (light gray), and the 808 buffering pairs (black). The BH-adjusted p-value (FDR) of 5% is marked. (**C**) Overlap of the 808 buffering genes with genes upregulated when the buffering partner is perturbed in K-562 cells (Replogle et al, 2022). For each CRISPRi perturbation in the transcriptome-wide dataset by Replogle and colleagues, gene expression signatures were z-score-transformed and to be sensitive, z > 2 was considered upregulated. The significance of the overlap was tested using a hypergeometric test. (**D**) Overlap of the 808 buffering gene pairs with genes showing high co-essentiality in DepMap. Z-score-transformed PCCs larger 4 were considered to show functional similarity. The significance of the overlap was tested using a hypergeometric test. (**E**) Sequence identity of Ensembl paralog pairs among the 808 high-quality buffering predictions (n = 227) and the pairs that were not predicted to buffer each other. (**F**) The density of high-quality buffering predictions of Ensembl paralogs by sequence identity bin. (**G**) Genomic location of paralog and non-paralog high-confidence buffering predictions. (**H**) Enrichment of both partners of a buffering pair on the same chromosome (syntenic buffering) of Ensembl paralog and non-paralog high-confidence buffering predictions. Enrichment was tested for significance using a hypergeometric test. (**I**) Number of buffered and buffering genes of the 808 predicted pairs on each chromosome. (**J**) Within and between-chromosome density of buffering pairs. Dot size corresponds to the change of the density of buffered and buffering genes compared to the global density. Significance was tested by hypergeometric testing with multiple hypothesis correction using the method by Benjamini-Hochberg (BH).

expression without correction or using multiple linear regression (Fig. 5A,B; Appendix Fig. S1I,L,N). Comparing the predictions with transcriptome-wide Perturb-Seq data (Replogle et al, 2022), we found that the expression of the buffering partner was 2.6-fold (p = 0.0027) more often upregulated in response to the perturbation of the buffered partner (Fig. 5C). While this mechanistically supports an active buffering mechanism for some predictions, this covered only 12 of the 808 predictions. Since buffering genes must fulfill similar functions to achieve buffering, we tested how the standard approach to predict functional similarity, namely DepMap co-essentiality estimates, covers the 808 buffering predictions. Again, while we observed a highly significant enrichment among our predictions for strong co-essentiality, only 21 pairs were covered (Fig. 5D). Finally, similar to previous studies (Esmaeili Anvar et al, 2024), paralog pairs with higher sequence similarity were more likely to buffer each other (Fig. 5E). However, only around 3% of Ensembl paralog pairs with high (>70%) sequence similarity were predicted as buffering (Fig. 5F). Together, this highlights the relation between our high-confidence list of gene buffering predictions and alternative datasets and methods and stresses a large number of unique observations.

Gene buffering can occur preferentially between different regions in the genome with therapeutic implications (Köferle et al, 2022). The 808 buffering predictions were located both within (syntenic) and between chromosomes (Fig. 5G). However, while buffering partners were generally not more likely to be syntenic buffering paralog gene pairs were 7 times less likely to be on the same chromosome (p = 6.4e-14; Fig. 5H). Overall, chromosomes harbor different numbers of buffering and buffered genes, with chr1 having the most buffered genes and chr11 showing the most buffering genes (Fig. 5I). The buffered-vs-buffering gene imbalance was also detectable for specific chromosome pairs. For instance, chrY genes often buffered chrX genes but not vice versa or chrX buffered chr1 or chr2 more often than expected (Fig. 5J). Together, the 808 high-quality buffering predictions covered all chromosomes but showed higher local rates for functional gene buffering.

## Discussion

We present BaCoN (<u>Ba</u>lanced <u>Co</u>rrelation <u>N</u>etwork), a method to correct correlation-based networks for systematically predicting

gene buffering from the Cancer Dependency Map (DepMap). BaCoN alone outperforms alternative methods and further increases the performance of predicting gene buffering when combined with a priori data normalization methods. By adding BaCoN to Cholesky whitening, we generate and characterize a high-quality set of 808 gene pair buffering predictions.

Supervised correction of confounding factors such as multiple linear regression represents the standard tool to systematically estimate gene buffering from DepMap gene effect and mRNA expression data (De Kegel et al, 2021; Pacini et al, 2024; Köferle et al, 2022). We show that cell lineage drives co-expression, which then leads to the identification of many false positive (FP) buffering genes, accounting for as much as 86.2% of the top 5000 predictions when only using PCC without normalization. However, taking into account lineage in multiple linear regression reduces this bias but still predicts 62% of genes from this likely FP category. BaCoN, constructed with the simple assumption that genes should be buffered by one or a few genes largely removed this bias in an unsupervised fashion. While similar network balancing strategies have been powerful in other types of molecular networks likely through removing biases (Bass et al, 2013; Billmann et al, 2016), this strategy matches the expected topology of genome-wide buffering networks.

Genomic location and proximity of genes have been a major source of potential FP fitness phenotypes in CRISPR-Cas9 screens and those biases are controlled well in recent DepMap Chronos score releases (Meyers et al, 2017; Lazar et al, 2024; Vinceti et al, 2024, 2023). We found that predicting gene buffering without normalization using Chronos scores with and without arm correction (AC) did not identify a relevant fraction of pairs with genomic proximity (< 1% of top 1000 predictions). However, only our top-performing methods BaCoN and Cholesky whitening increased the number of unwanted proximal pairs substantially (15–25% of top 1000 predictions) using uncorrected Chronos scores and benefited substantially from corrections. This suggests that additional corrections, which are routinely applied to Chronos scores, are important for improved prediction of gene buffering.

BaCoN changes prediction score distributions by balancing the strongest predictions in a way that genes with many predicted partners are weakened whereas genes with fewer predicted partners are boosted. This specific post hoc correction of correlation-based networks only acts on gene pairs with a high level of covariation. In

contrast, a priori data normalization methods such as data whitening can remove major axes of variation of the data and emphasize covariation between genes pairs not seen in the original data, which can lead to strongly remodeled prediction score distributions even pinpointing gene pairs with negative correlation in the original data. This is what we coined unlinking of the original input gene effect and expression data and suggest to interpret the resulting scores with caution. When combining Cholesky whitening with BaCoN various metrics showed that the scores outperformed alternative methods, allowing us to define a set of 808 high-confidence gene buffering predictions. This list includes clinically relevant predictions of *YAP1* buffering its paralog *WWTR1* and *WWTR1* buffering *YAP1*, or *ARID1A* buffering its paralog *ARID1B* and vice versa (Frost et al, 2023; Helming et al, 2014). Notably, all of those four gene pairs displayed a PCC between a z-score of $-1$ and 1 in the original data. Moreover, we were able to experimentally validate unknown buffering predictions with low correlation in the uncorrected data. For instance, we found buffering between ubiquitin-conjugating enzymes UBE2C and UBE2S, which promote mitotic cell cycle progression via the anaphase-promoting complex. In accordance with our findings both have been reported to be synthetically viable with the spindle assembly checkpoint, which inhibits the anaphase-promoting complex (Wild et al, 2016). Together, this emphasizes that our list of high-confidence predictions pinpoint important biological buffering mechanisms that cannot be derived from the DepMap otherwise.

The DepMap has consistently added more cell models and intends to expand past the currently approximately 1100 cell lines targeting more than 10,000 cancer models (Boehm et al, 2021). Given that BaCoN particularly excels with an increasing number of screens and genes tested, this suggests that BaCoN can substantially help the search for gene buffering going forward. Notably, while the most comprehensively represented cancer entity in the DepMap counts just over 100 screens, the anticipated increasing number of cell models will likely give rise to tissue-specific, BaCoN-supported gene buffering predictions.

Together, we present BaCoN, a new method to balance correlation networks that use orthogonal inputs such as expression and perturbation data to predict functional buffering between genes without prior knowledge of experimental confounding factors. BaCoN can be combined with existing normalization methods and gains predictive power with increasing numbers of DepMap screens.

## Methods

### Reagents and tools table

| Reagent/resource | Reference or source | Identifier or catalog number |
|---|---|---|
| **Experimental models** | | |
| hTERT-RPE1 | Gifted by Holland Lab | |
| **Recombinant DNA** | | |
| pLenti_mpx_SPCas9_h7SK (PacI)_U6 (I-SceI)_PGK_Puro_v2 | Addgene | 189632 |
| **Oligonucleotides and other sequence-based reagents** | | |
| gRNA oligos | Henkel et al, 2020 | Dataset_EV4.xlsx |

| Reagent/resource | Reference or source | Identifier or catalog number |
|---|---|---|
| **Chemicals, enzymes, and other reagents** | | |
| psPAX2 | Addgene | 12260 |
| pMD2.G | Addgene | 12259 |
| Opti-MEM | ThermoFisher Scientific | 31985062 |
| GeneJuice | Sigma-Aldrich | 70967-4 |
| DMEM/F12 | ThermoFisher Scientific | 11320-074 |
| Hygromycin B | Capricorn Scientific | HYG-H |
| Puromycin | InvivoGen | ant-pr-5 |
| Polybrene | Sigma-Aldrich | H9268 |
| **Software** | | |
| BD FACS Diva™ v9.0 | BD Biosciences | |
| R 4.4.2 | https://www.r-project.org/ | |
| biomaRt 2.58.0 | Data request on 15.04.2024 (Durinck et al, 2009) | |
| BSgenome.Hsapiens.UCSC.hg38 1.4.5 | 10.18129/B9.bioc. BSgenome.Hsapiens. UCSC.hg38 | |
| sva 3.50.0 | Leek et al, 2012 | |
| UCSC.utils 1.2.0 | 10.18129/ B9.bioc.UCSC.utils | |
| whitening 1.4.0 | Kessy et al, 2018 | |
| **Other** | | |
| BD FACSCanto™ | BD Biosciences | 338962 |

### NCBI Refseq gene database

Information on the genomic location of genes was retrieved from the NCBI Refseq database (Pruitt et al, 2009) (Homo sapiens, version GCF_000001405.40_GRCh38.p14). The database was filtered for protein-coding genes. Genes were mapped to their respective chromosome arm using UCSC Hg38 cytoband data (database version 2022-10-28). The information was retrieved using the UCSC.utils R package (version 1.2.0).

### Gene effect and expression data

From the DepMap 23Q2 version, CCLE gene expression data (http://depmap.org/portal/download/all/?releasename=DepMap +Public+23Q2&filename=OmicsExpressionProteinCodingGenes TPMLogp1.csv) as well as Chronos gene fitness effect scores (http:// depmap.org/portal/download/all/?releasename=DepMap+Public +23Q2&filename=CRISPRGeneEffect.csv) were imported. Missing Chronos scores were imputed as gene-wise score means. Genes with a mean expression of less than 1 were removed from the CCLE expression dataset, leaving 12,401 genes in the matrix. The Chronos score matrix was restricted to 11,885 genes that made the gene expression threshold applied to the CCLE data. To allow the computation of association indices, where fitness effects and gene expression were aligned in every cell line, the datasets were reduced to the intersecting 1019 cell lines. Unless stated otherwise, these datasets were used to compute buffering predictions. Cell line

metadata on cancer models (http://depmap.org/portal/download/all/?releasename=DepMap+Public+23Q2&filename=Model.csv) as well as assay conditions (http://depmap.org/portal/download/all/?releasename=DepMap+Public+23Q2&filename=ModelCondition.csv) was retrieved from the same DepMap version. A list of cell line lineages was defined using OncotreePrimaryDisease for the lineages lung, skin as well as peripheral nervous system, OncotreeLineage classification was used for all other cell lines.

## Additional datasets

### Methylation
Data on methylation profiles (http://depmap.org/portal/data_page/?tab=allData&releasename=Methylation%20(RRBS)&filename=CCLE_RRBS_TSS_1kb_20180614.txt), generated by the Broad institute, was downloaded from the DepMap portal (https://doi.org/10.1038/nature15736). If genes were profiled multiple times in the same cell line, their measurements were mean-summarized to receive one value per gene and cell line. The resulting dataset contains methylation data for 12,898, measured in 839 cell lines.

### Copy number variation
Copy number variation data was imported from the 24Q2 DepMap portal (http://depmap.org/portal/data_page/?tab=allData&releasename=DepMap%20Public%2024Q2&filename=OmicsCNGene.csv). The 24,383 genes were subsetted to the 12,174 protein-coding genes of the 24Q2 gene expression object, and the 1066 intersecting cell lines with the 24Q2 Chronos score object were chosen.

### Demeter dependency scores
DEMETER2 scores from Project Achilles were imported from the DepMap portal (http://depmap.org/portal/data_page/?tab=allData&releasename=DEMETER2%20Data%20v6&filename=D2_Achilles_gene_dep_scores.csv). To ensure comparability between the datasets, the genes were subsetted for the ones also represented in the gene expression data.

### Protein expression
Nusinow and colleagues provided mass-spectrometry-derived protein expression data for a large fraction of genes and cell lines (Nusinow et al, 2020). We imported these data from the DepMap portal (http://depmap.org/portal/data_page/?tab=allData&releasename=Proteomics&filename=protein_quant_current_normalized.csv). Expression values for genes that were measured multiple times in one cell were mean-summarized per gene. We reduced the number of genes to the ones measured in at least 200 cell lines. The resulting 8689 genes, measured across 378 cell lines, were used as an alternative to gene expression data.

### Expression including protein-coding and non-coding RNAs
To predict buffering interactions between genes and non-coding RNAs (ncRNAs), we expanded our gene expression dataset with an extended matrix including data on ncRNAs, available for the 24Q2 Depmap release (http://depmap.org/portal/data_page/?tab=allData&releasename=DepMap%20Public%2024Q2&filename=OmicsExpressionAllGenesTPMLogp1Profile.csv). We used the gene database to remove entries from the gene expression object that were not listed as either protein-coding gene, lncRNA, miRNA, snRNA or ncRNA. Protein-coding genes were filtered by the same criteria used in all the analysis

throughout his paper (with the exception of the analysis presented in Fig. 4). Non-coding (nc)RNAs expressed at >1 TPM in at least 100 cell lines. The resulting expression object contained 11,459 entries, consisting of 10,115 protein-coding genes and 1344 ncRNAs. The genes in the Chronos gene effect matrix were reduced to the 9965 genes also included in the gene expression object. The intersecting 1066 cell lines between the objects were used to compute a PCC + BaCoN matrix on the Cholesky-whitened datasets. We used the weakest BaCoN score of our high-confidence prediction set as a cutoff and extracted the top-scoring 55 predicted pairs between a ncRNA and a protein-coding gene.

## Defining paralog pairs to evaluate gene buffering predictions

Considering the lack of a comprehensive gold standard set of buffering gene pairs, we assembled a collection of gene pair sets, consisting of human protein-coding Ensembl paralogs and Ohnologs, as well as multiple sets of predicted and/or experimentally validated SL paralog pairs.

### Ensembl paralogs
A set of 191,582 protein-coding paralog pairs was obtained from the Homo sapiens Ensembl database (Yates et al, 2019), using the BioMart R package (version 2.58.0, data request on 15.04.2024) (Durinck et al, 2009). The database request was conducted using the filters "with_hsapiens_paralog" = TRUE and transcript biotype = "protein coding" to select protein-coding pairs of human paralogs. The resulting list of paralog pairs was filtered for a minimum sequence identity of 20%.

### Ohnologs
A set of Ohnolog gene pairs was built based on the Curie database (http://ohnologs.curie.fr/cgi-bin/DownloadBrowse.cgi?crit=%5BC%5D&org=hsapiens&opt=pairs&wgd=2R) (Singh and Isambert, 2019), using relaxed filter criteria (outgroup q-score <0.05 and self-comparison q-score < 0.3). After filtering for protein-coding genes, the remaining set contained 7328 pairs (Appendix Fig. S2E).

### SL predicted by De Kegel and colleagues
We obtained two sets of paralog pairs from De Kegel et al, 2021 (De Kegel et al, 2021). 12 experimentally validated SL paralog pairs (http://www.cell.com/cms/10.1016/j.cels.2021.08.006/attachment/aa048948-5014-471a-89cf-16b83f522014/mmc2.xlsx) were defined as 'De Kegel validated SL' (Appendix Fig. S2F). 131 SL paralog predictions defined by a negative A2_status coefficient at 10% FDR were defined as 'De Kegel SL' (http://www.cell.com/cms/10.1016/j.cels.2021.08.006/attachment/8c1dd4f0-6921-4680-a482-5f8199bbb77c/mmc6.csv).

### SL pairs defined by Anvar and colleagues
We used two sets of SL paralog pairs based on the work by Anvar and colleagues (Esmaeili Anvar et al, 2024). The authors conducted a meta-analysis on five studies that conducted multiplex synthetic lethality screens on paralog pairs, using different Cas12-based systems (Dede et al, 2020; Gonatopoulos-Pournatzis et al, 2020; Thompson et al, 2021; Ito et al, 2021; Parrish et al, 2021). By re-scoring all pairs tested in these studies, the authors had generated a

standard set of 388 paralog pairs. We refer to this set of gene pairs as 'Anvar et al SL standard'.

The authors also conducted four screens using two CRISPR/Cas12a multiplex libraries (a prototype library targeting about 2000 paralog families in K-562 and A549 cells, as well as the Inzolia library on A375 and MEL-JUSO cells, targeting about 4000 paralog pairs, triples, and quads). In accordance with their approach, we computed the interaction strength (Delta Log Fold Change, dLFC) as the difference between the observed double-knockout LFC of a paralog pair and the sum of single-knock-out LFCs. Pairs with a strong negative fitness phenotype (dLFC < −1) in at least one of the screens were defined as hits. We discarded paralog families with more than two genes and used the remaining set of 517 SL paralog pairs as standard. We refer to this set of gene pairs as 'Anvar SL' (Appendix Fig. S2G).

### SL gene pairs by Ito and colleagues

A set of 1829 paralog pairs was generated based on Gemini scores from work by Ito and colleagues (Ito et al, 2021). We used relaxed filter criteria and required a paralog pair only to show a positive Gemini LFC in at least one of the eleven tested cell lines (FDR <5%) to be classified as SL (Appendix Fig. S2G).

### SL gene pairs by Thompson and colleagues

Thompson and colleagues performed combinatorial CRISPR screens co-perturbing paralog pairs in A375, MeWo and RPE-1 cell lines (Thompson et al, 2021). The 27 paralog pairs that were identified as SL in more than one cell line were obtained from this study (Appendix Fig. S2G).

### Diversity index

We observed that a small number of genes dominated the top PCC buffering predictions, likely due to confounding with tissue-specific co-expression (Fig. 1E–G; Appendix Fig. S3A,B). This observation led to the conclusion that the low diversity of a prediction set (a small number of genes contributing to a large number of gene pairs) can be used as a marker for biased predictions, and that quantifying the diversity of a prediction set is a useful quality characteristic to compare the predictions made using different correction methods. The diversity index reflects the minimal number of genes contributing to 50% of the pairs in a set of predictions (Appendix Fig. S3C,D). For very diverse prediction sets (consisting of many independently paired genes) the coefficient is 1. It is computed by first computing the maximum possible number of genes forming 50% of the tested pairs (e.g., 100 genes for a set of 100 predicted pairs). After that, the number of times each gene is observed, starting with the most frequent one, are cumulatively added, until the maximum possible number of genes is reached. The resulting value is divided by the maximum possible number.

### Proximity enrichments

The genomic position of each gene was approximated as the mean between start and end coordinate. For each gene pair with both partners located on the same chromosome, the genomic distance was computed as the absolute difference between the genomic positions of both genes.

For a set of buffering predictions, enrichments for different proximity windows were computed using hypergeometric tests, comparing the density of predicted pairs within the respective proximity window among all buffering predictions of the set against the global density of pairs in this proximity window among all 147 million tested pairs. Based on observations of extreme enrichments of predictions with genomic distances <10 million base pairs (Appendix Fig. S3), these pairs were flagged as proximal pairs and treated as false positives.

### Empirical false discovery rate based on defined FP and TP gene pairs

Based on our observations on co-expression between buffering genes as well as proximity between the buffering and the buffered genes being indicators of false positive predictions, we computed a false discovery rate for each prediction. First, predictions that were either Ensembl paralogs, Ohnologs, part of the Anvar, Anvar standard, Ito or Thompson et al SL sets or self-addictions were used as true positive (TP) predictions. A gene pair was flagged as false positive (FP) if it is no TP and fulfills at least one of the following criteria: (1) The buffering and buffered gene are on the same chromosome arm and less than 10 Mbp apart, reflecting the proximity-biased predictions. (2) If a buffered gene has more than one buffering partner, the co-expression between all of these buffering genes is computed. Within this cluster, every pair is designated as FP if the buffering gene is co-expressed with a higher-rank buffering gene ($z > 3$). Using this definition of TP and FP pairs, the FDR is computed as $FP/(FP + TP)$.

### Pearson's and Spearman's rank correlation coefficients

To quantify the buffering capacity between a gene pair ($G_A$ - $G_B$), the association between the CCLE expression of $G_A$ and the Chronos scores of $G_B$ was quantified. The Pearson's correlation coefficient (PCC) was used as the default association index, using the cor() function in R on the Chronos and gene expression matrix, after aligning the cell lines of both objects to the intersecting 1019. The result is a correlation matrix with dimensions $12,401 \times 11,885$, containing coefficients for each tested gene pair. The Spearman's rank correlation coefficient (SCC) was computed analogously.

### Multiple linear regression-based gene-gene association

The problem of quantifying the association strength of variables while accounting for the effects of known covariates can be addressed using multiple linear regression. For each gene pair of the respective expression and Chronos matrices, we trained a linear model, using the cell line covariates lineage as well as growth pattern (suspension, adherent) as independent variables. For each gene pair $G_A$-$G_B$, the dependency of the Chronos scores of $G_B$ on $G_A$ mRNA expression was computed. The following formulas were tested independently: $G_B$ Chronos ~ $G_A$ Expression, $G_B$ Chronos ~ $G_A$ Expression + Lineage, as well as $G_B$ Chronos ~ $G_A$ Expression + Growth pattern. The predictions were ranked using the negative logarithm of the main predictor variable ($G_A$ Expression) $P$ value as an association metric. As the predictor $P$ value scales independently from the slope of the model, we inverted this index for pairs with a

negative slope. This allowed to distinguish between negatively and positively associated pairs. The output of gene pair buffering predictions using linear models is a matrix of negative logarithmic $P$ values, with the highest positive values describing the strongest association signals.

## Batch correction using ComBat

Batch effects are a common source of confounding in large, multi-experimental datasets, complicating direct comparison of samples of different origin and experimental conditions (Johnson et al, 2007). A well-established method to adjust batch effects in microarray data is a Bayesian framework implemented in the ComBat algorithm. Using the ComBat function from the R sva package (version 3.50.0, (Leek et al, 2012)), we adjusted for batch effects within the input datasets. This was done independently for the gene expression data as well as the fitness effects. After that, a correlation matrix was generated from the adjusted data and used to quantify the buffering association strength. As batch information, the cell line lineages of origin as well as on adherent or suspension growth pattern during the cell culture were used along the cell line dimension. When correcting batches along the gene dimension of a matrix, chromosome as well as chromosome arm location were used.

## Data whitening

Data whitening describes data transformations that normalize covariance and variance (Gheorghe and Hart, 2022). Several methods of data whitening are established, and whitening based on Cholesky decomposition (Cholesky whitening) has been successfully demonstrated to remove bias in co-essentiality networks (Wainberg et al, 2021). We independently tested two whitening methods, namely, Cholesky whitening, as well as whitening based on principal component analysis (PCA-whitening). Using the whitening R package (version 1.4.0) (Kessy et al, 2018), both the expression as well as the Chronos data were whitened prior to computing the PCC-correlation matrix. To normalize the covariance between cell lines but not genes, we transposed the input objects prior to whitening and then re-transposed the whitened matrices before computing the correlation matrix.

## BaCoN

BaCoN is deployed on a correlation matrix with the dimensions $n \times m$, with $n$ describing the number of tested genes from the expression dataset and $m$ defining the number of genes with Chronos scores. BaCoN balances the correlation of each gene pair $AB$ of a matrix based on the fraction of more extreme hits of the genes of the respective pair. For each gene pair $AB$ of the correlation matrix with $r_{AB} \geq 0$, first a correction factor $k$ is subtracted from the correlation coefficient $r_{AB}$. The correction factor $k$ can range from 0 to 1, where 0 transforms the network the strongest. We chose $k$ at 0.05 so BaCoN (i) detected a high number of self-addictions among the top predictions and (ii) removed co-expression between buffering partners (Appendix Fig. S2A,D). At the same time, we aimed to reduce ties among BaCoN scores and favored a moderate data transformation (Appendix Fig. S1B,C).

After subtracting $k$ is subtracted from the correlation coefficient $r_{AB}$, the density of all adjacent and more extreme correlation scores is calculated and subtracted from 1, resulting in high BaCoN scores for gene pairs with few more extreme partners:

$$BaCoN_{AB_{pos}} = 1 - \frac{\sum_{i=1}^{n} r_{AN} > r_{AB} - k + \sum_{j=1}^{m} r_{MB} > r_{AB} - k}{n + m}$$

For each pair $AB$ with a negative correlation ($r_{AB} < 0$), BaCoN scores for negatively correlating gene pairs are generated analogously, with inverted signs. The correction factor $k$ is added to $r_{AB}$, and the density of lower correlations of the pair is added to $-1$:

$$BaCoN_{AB_{neg}} = -1 + \frac{\sum_{i=1}^{n} r_{AN} < r_{AB} + k - \sum_{j=1}^{m} r_{MB} < r_{AB} + k}{n + m}$$

The result is a matrix with the same dimensions as the input matrix.

## Random subsets

Reliable predictions on gene buffering require a sufficient number of cell lines to associate the expression of one gene with the fitness scores of another gene. We quantified the impact of varying the number of cell lines used on predicting buffering to estimate the impact of BaCoN on the future DepMap and to benchmark the buffering prediction capacity of the best-performing methods (PCC, PCC + BaCoN, ComBat (tissue) + PCC, ComBat (tissue) + PCC + BaCoN, multiple linear regression with tissue as covariate). The 1019 overlapping cell lines between the gene expression and Chronos matrices were randomly sub-sampled. Sets of the sizes 100, 200 as well as 500 were generated, with 10 replicates each. We used the sub-sampled expression and Chronos matrices to generate association matrices using the previously described methods, keeping the number of tested gene pairs constant.

## Dynamic gene space

Conceptually, we expect that buffering predictions require the buffering partner to be expressed in a sufficient number of cell lines and the knockout of the buffered partner to have a certain impact on cell fitness. To test how removing genes with low expression or essentiality signals influences predictions, we used two types of thresholds that were applied to the gene expression and chronos score data, respectively. The expression thresholds (10, 30, 60, 100, 300, 600, 900, 1000) were used to remove genes that do not show an expression (log2 TPM + 1) of $\geq 3$ in at least the respective number of cell lines, removing genes with low expression signal. The Chronos thresholds ($-0.2, -0.25, -0.3, -0.35, -0.4, -0.45, -0.5, -0.6, -0.8, -1.0, -1.2$ and $-1.5$) were used to select only genes that did show the respective essentiality in a minimum of 30 cell lines. This way, genes with low essentiality across most of the cell lines were removed. Each of the 8 subset gene expression matrices was combined with each of the 12 subset Chronos matrices, resulting in a total of 96 prediction matrices of different dimensions. The resulting matrices were used to predict buffering, using the overlapping 1019 cell lines. The number of gene pairs in the respective universe was computed as the product of the number of all genes in the expression data (potential buffering) and all potentially buffered genes (Chronos scores) that were not filtered

by applying the filters. Using the previously described reference gene pair sets, the paralog density was computed as the number of gene pair hits divided by the number of total gene pairs of a matrix. The paralog enrichment was computed as the density of paralog hits among the top 100 predictions, divided by the paralog density (see https://github.com/billmannlab/BaCoN_manuscript for details).

### High-confidence buffering gene pair set

We aimed to generate a larger set of high-quality buffering predictions (Fig. 5). Based on our finding that the top 1000 predictions generated by combining Cholesky whitening with BaCoN outperformed the tested competing methods in terms of paralog prediction performance as well as prediction diversity and the metric-summarizing empirical FDR and this trend continued past those top 1000 predictions (Fig. 3; Appendix Figs. S4 and S5), we based a high-quality buffering prediction set on this set of gene pairs. As described above, we computed a PCC-based correlation matrix from the Cholesky-whitened fitness effect as well as the Cholesky-whitened gene expression data. BaCoN was then applied on the resulting PCC matrix. Overall, we nominated 1245 gene pairs at a metric-summarizing empirical FDR of 50%. This ensured that pairs not assigned to the TP buffering prediction set were called at this FDR. After filtering self-addicted gene pairs (pairs with matching buffering and buffered gene), genomically proximal genes, as well as pairs where the buffering genes are co-expressed with another gene buffering the same partner (as described in "FDR"), 808 high-confidence gene pairs remained in the set.

### Experimental validation of predicted buffering gene pairs in cell culture

To experimentally validate predictions of genetic interactions, we performed FACS-based FP-competition assays comparing cell fitness after depleting single or combinations of paralogs. For each gene combination, we selected SpCas9 knockout gRNAs from an empirically validated genome-wide CRISPRko library and cloned the corresponding gRNA sequences into the previously reported vector pLenti_mpx_SpCas9_h7SK(PacI)_U6(I-SceI)_PGK-puro_v2 (Addgene: 189632) (Wegner et al, 2020; Henkel et al, 2020) (Dataset EV4). For lentivirus production, 3.3 µg of the respective gRNA-containing construct, 2.7 µg of psPAX2 (Addgene: 12260) and 1 µg of pMD2.G (Addgene:12259) were mixed with 200 µl ml of Opti-MEM (ThermoFisher Scientific, 31985062) and 21 µl of GeneJuice transfection reagent (Sigma-Aldrich, 70967-4) and dropwise added to HEK293T cells. Virtual particles were harvested 48 h after transfection and stored at $-80\,°C$. For the GFP competition assay, puromycin-sensitive hTERT-RPE1 cells were transduced with the respective KO constructs while hTERT-RPE1_GFP positive cells were transduced with an AAVS1 targeting control construct. The next day, the cells were selected with puromycin and 3 days after the transduction, the cells were mixed with a target distribution of 80% (WT) to 20% (GFP-positive cells) and seeded in 6-well plates as triplicates in DMEM Nutrient Mixture F-12 media (DMEM/F12, ThermoFisher Scientific, 11320-074), supplemented with 10% FBS, 1% penicillin-streptomycin and 0.01 mg/ml hygromycin B (Capricorn Scientific, HYG-H) and incubated at $37\,°C$ and 5% $CO_2$. The percentage of WT and GFP

cells for every sample was recorded with a BD FACS Diva from the initial day of mixing until day 14 after transduction. The Log2-fold changes (LFCs) for every sample and day were calculated with day 0 as a reference. Genetic interactions were determined by computing the difference between the observed combinatorial phenotype and comparing the sum to the sum of the LFCs of the respective single phenotypes per gene pair.

## Data availability

Reproducibility of this manuscript: the documented code used to produce all analyses presented in this manuscript can be obtained from https://github.com/billmannlab/BaCoN_manuscript. Use BaCoN: The BaCoN R package can be obtained from https://github.com/billmannlab/BaCoN/. A Python command line implementation can be obtained from the same repository or https://github.com/billmannlab/pyBaCoN/.

The source data of this paper are collected in the following database record: biostudies:S-SCDT-10_1038-S44320-025-00103-7.

## Peer review information

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

## Acknowledgements

The authors thank all members from the Billmann lab and the Institute of Human Genetics at the University of Bonn and University Hospital Bonn for helpful discussions. The authors thank Chad Myers for helpful discussions. MB was funded by the German Research Foundation (DFG; TRR259, grant number 397484323).

## Author contributions

**Thomas Rohde**: Conceptualization; Data curation; Software; Formal analysis; Investigation; Visualization; Methodology; Writing—original draft; Writing—review and editing. **Talip Yasir Demirtas**: Software. **Sebastian Süsser**: Validation. **Angela Helen Shaw**: Investigation. **Manuel Kaulich**: Validation. **Maximilian Billmann**: Conceptualization; Resources; Data curation; Software; Formal analysis; Supervision; Funding acquisition; Investigation; Visualization; Methodology; Writing—original draft; Project administration; Writing—review and editing.

Source data underlying figure panels in this paper may have individual authorship assigned. Where available, figure panel/source data authorship is listed in the following database record: biostudies:S-SCDT-10_1038-S44320-025-00103-7.

## Funding

## Disclosure and competing interests statement

The authors declare no competing interests.

