## [Peer Review File · Molecular Systems Biology]

BaCoN (Balanced Correlation Network) improves prediction of gene buffering

Thomas Rohde, Talip Demirtas, Sebastian Suesser, Angela Shaw, Manuel Kaulich, and Maximilian Billmann

Corresponding author(s): Maximilian Billmann (m.billmann@uni-bonn.de)

Review Timeline:	Submission Date:	22nd Jul 24
	Editorial Decision:	11th Sep 24
	Revision Received:	21st Feb 25
	Editorial Decision:	17th Mar 25
	Revision Received:	3rd Apr 25
	Accepted:	7th Apr 25

Editor: Jingyi Hou

Transaction Report:

11th Sep 2024

Manuscript Number: MSB-2024-12533

Title: BaCoN (Balanced Correlation Network) improves prediction of gene buffering

Author: Maximilian Billmann

Thomas Rohde

Talip Demirtas

Angela Shaw

Dear Dr. Billmann,

Thank you for submitting your work to Molecular Systems Biology. We have now heard back from the three reviewers who agreed to evaluate your manuscript. As you will see from the reports below, the reviewers think the study potentially interesting. However, they raised a series of concerns, which we would ask you to address in a major revision.

The reviewers' recommendations are relatively clear, so there is no need to reiterate the points listed below. All the issues raised by the reviewers need to be satisfactorily addressed. As you may already know, our editorial policy allows in principle a single round of major revision, and it is therefore essential to provide responses to the reviewers' comments that are as complete as possible.

On a more editorial level, we would ask you to address the following issues:

- Please provide a .docx formatted version of the manuscript text (including legends for main figures, EV figures and tables). Please make sure that the changes are highlighted to be clearly visible.
- Please provide individual production quality figure files as .eps, .tif, .jpg (one file per figure).
- Please provide a .docx formatted letter INCLUDING the reviewers' reports and your detailed point-by-point responses to their comments. As part of the EMBO Press transparent editorial process, the point-by-point response is part of the Review Process File (RPF), which will be published alongside your paper.
- Please note that all corresponding authors are required to supply an ORCID ID for their name upon submission of a revised manuscript.
- We replaced Supplementary Information with Expanded View (EV) Figures and Tables that are collapsible/expandable online (see examples in <http://msb.embopress.org/content/11/6/812>). A maximum of 5 EV Figures can be typeset. EV Figures should be cited as 'Figure EV1, Figure EV2' etc... in the text and their respective legends should be included in the main text after the legends of regular figures.

Additional Tables/Datasets should be labeled and referred to as Table EV1, Dataset EV1, etc. Legends have to be provided in a separate tab in case of .xls files. Alternatively, the legend can be supplied as a separate text file (README) and zipped together with the Table/Dataset file.

For the figures and tables that you do NOT wish to display as Expanded View figures, they should be bundled together with their legends in a single PDF file called *Appendix*, which should start with a short Table of Content. Each legend should be below the corresponding Figure/Table in the Appendix. Appendix figures and tables should be referred to in the main text as: "Appendix Figure S1, Appendix Figure S2, Appendix Table S1" etc. See detailed instructions regarding expanded view here: <https://www.embopress.org/page/journal/17444292/authorguide#expandedview>.

- Before submitting your revision, primary datasets (and computer code, where appropriate) produced in this study need to be deposited in an appropriate public database (see <http://msb.embopress.org/authorguide-dataavailability> <https://www.embopress.org/page/journal/17444292/authorguide#dataavailability>). Please remember to provide a reviewer password if the datasets are not yet public. The accession numbers and database should be listed in a formal "Data Availability" section (placed after Materials & Method) that follows the model below (see also <https://www.embopress.org/page/journal/17444292/authorguide#dataavailability>). Please note that the Data Availability Section is restricted to new primary data that are part of this study.

Data availability

-At EMBO Press we ask authors to provide source data for the main figures. Our source data coordinator will contact you to discuss which figure panels we would need source data for and will also provide you with helpful tips on how to upload and organize the files.

- Our journal encourages inclusion of *data citations in the reference list* to directly cite datasets that were re-used and obtained from public databases. Data citations in the article text are distinct from normal bibliographical citations and should directly link to the database records from which the data can be accessed. In the main text, data citations are formatted as follows: "Data ref: Smith et al, 2001". In the Reference list, data citations must be labeled with "[DATASET]". A data reference must provide the database name, accession number/identifiers and a resolvable link to the landing page from which the data can be accessed at the end of the reference. Further instructions are available at .

- We updated our journal's competing interests policy in January 2022 and request authors to consider both actual and perceived competing interests. Please review the policy <https://www.embopress.org/competing-interests> and update your competing interests if necessary. Please use the heading "Disclosure statement and competing interests".

- All Materials and Methods need to be described in the main text using our 'Structured Methods' format. According to this format, the Methods section includes a Reagents and Tools Table (listing key reagents, experimental models, software and relevant equipment and including their sources and relevant identifiers) followed by a Methods and Protocols section describing the methods, ideally using a step-by-step protocol format. The aim is to facilitate adoption of the methodologies across labs. Please download and fill our Reagents and Tools Table template (.docx), which you can find in our author guidelines: <https://www.embopress.org/page/journal/17444292/authorguide#structuredmethods>.

An example of a Method paper with Structured Methods can be found here: <https://www.embopress.org/doi/10.15252/msb.20178071>.

-Regarding data quantification:

Please ensure to specify the name of the statistical test used to generate error bars and P values, the number (n) of independent experiments (please specify technical or biological replicates) underlying each data point and the test used to calculate p-values in each figure legend. Discussion of statistical methodology can be reported in the materials and methods section, but figure legends should contain a basic description of n, P and the test applied. Graphs must include a description of the bars and the error bars (s.d., s.e.m.). Please also include scale bars in all microscopy images.

- Please provide a "standfirst text" summarizing the study in one or two sentences (approximately 250 characters, including space), three to four "bullet points" highlighting the main findings and a "synopsis image" (550px width and 400-600 px height, PNG format) to highlight the paper on our homepage.

Here are a couple of examples:

<https://www.embopress.org/doi/10.15252/msb.20199356>

<https://www.embopress.org/doi/10.15252/msb.20209475>

<https://www.embopress.org/doi/10.15252/msb.209495>

When you resubmit your manuscript, please download our CHECKLIST (<https://www.embopress.org/pb-assets/embo-site/EMBO%20Press%20Author%20Checklist-1642513524327.xlsx>) and include the completed form in your submission.

Please note that the Author Checklist will be published alongside the paper as part of the transparent process (<https://www.embopress.org/page/journal/17444292/authorguide#transparentprocess>).

If you feel you can satisfactorily deal with these points and those listed by the referees, you may wish to submit a revised version of your manuscript. Please attach a covering letter giving details of the way in which you have handled each of the points raised by the referees. A revised manuscript will be once again subject to review and you probably understand that we can give you no guarantee at this stage that the eventual outcome will be favorable.

I look forward to receiving your revised manuscript soon.

Kind regards,
Jingyi

Jingyi Hou, PhD
Scientific Editor
Molecular Systems Biology

We realize that it is difficult to revise to a specific deadline. In the interest of protecting the conceptual advance provided by the work, we recommend a revision within 3 months (10th Dec 2024). Please discuss the revision progress ahead of this time with the editor if you require more time to complete the revisions. Use the link below to submit your revision:

IMPORTANT: When you send your revision, we will require the following items:

1. the manuscript text in LaTeX, RTF or MS Word format
2. a letter with a detailed description of the changes made in response to the referees. Please specify clearly the exact places in the text (pages and paragraphs) where each change has been made in response to each specific comment given
3. three to four 'bullet points' highlighting the main findings of your study
4. a short 'blurb' text summarizing in two sentences the study (max. 250 characters)
5. a 'thumbnail image' (550px width and max 400px height, Illustrator, PowerPoint or jpeg format), which can be used as 'visual title' for the synopsis section of your paper.
6. Please include an author contributions statement after the Acknowledgements section (see <https://www.embopress.org/page/journal/17444292/authorguide>)

7. Please complete the CHECKLIST available at (<https://bit.ly/EMBOPressAuthorChecklist>).

Please note that the Author Checklist will be published alongside the paper as part of the transparent process (<https://www.embopress.org/page/journal/17444292/authorguide#transparentprocess>).

See also figure legend guidelines: <https://www.embopress.org/page/journal/17444292/authorguide#figureformat>

9. Please note that corresponding authors are required to supply an ORCID ID for their name upon submission of a revised manuscript (EMBO Press signed a joint statement to encourage ORCID adoption).

(<https://www.embopress.org/page/journal/17444292/authorguide#editorialprocess>)

Currently, our records indicate that the ORCID for your account is 0000-0002-6556-9594.

Link Not Available

11. Include a Reagents and Tools Table as part of the Methods section, which can be downloaded from our author guidelines (<https://www.embopress.org/page/journal/17444292/authorguide#structuredmethods>)

*** PLEASE NOTE *** As part of the EMBO Press transparent editorial process initiative (see our Editorial at <https://dx.doi.org/10.1038/msb.2010.72>), Molecular Systems Biology publishes online a Review Process File with each accepted manuscripts. This file will be published in conjunction with your paper and will include the anonymous referee reports, your point-by-point response and all pertinent correspondence relating to the manuscript. If you do NOT want this File to be published, please inform the editorial office at msb@embo.org within 14 days upon receipt of the present letter.

Reviewer #1:

The authors present BaCoN: a new computational method that outperforms existing correlation-based approaches for predicting buffering gene pairs, i.e., pairs of genes where the loss of function in one gene (due to low expression or a genomic alteration) creates a selective dependency on the second gene for cellular survival.

The authors demonstrate the effectiveness of their tool by reanalyzing publicly available data from the Cancer Dependency Map (DepMap) repository. Specifically, they use two data matrices from over 1000 cancer cell lines, capturing genome-wide expression profiles (via bulk RNA sequencing) and genetic dependencies (in terms of viability reduction upon CRISPR-Cas9 targeting). They employ a series of robust metrics, such as the number of recovered paralog-gene pairs among the top hits, to

validate BaCoN's performance.

In brief, BaCoN accounts for the background covariability of each gene and post-normalizes correlation scores based on the overall correlation of a gene's expression in relation to the fitness profiles of all other genes. This approach is simple, yet elegant and effective, and it offers a valuable contribution to the identification of gene-buffering pairs from large-scale DepMap datasets. This tool will be of interest to researchers working in the field.

However, there are several points I would like to see addressed before considering this manuscript for publication.

First, the authors argue that because buffering predictions align two independent data matrices—gene perturbation fitness effects and mRNA expression—the independent, a priori normalization of these data might unlink biologically meaningful covariations. This reasoning is poorly justified and should be supported by more data-driven observations. Are there any examples of meaningful covariations that are lost when the two data matrices are processed separately but preserved through post-hoc normalization?

Additionally, the normalization method should be tested more comprehensively. For instance, are self-addictions (i.e., the tendency for a gene's expression to predict dependency on that gene) better detected after BaCoN correction compared to other methods that integrate mRNA expression and genetic dependency?

While the authors discuss factors that can obscure the interpretation of data from CRISPR-Cas9 screens (such as CN-amplification), they should also mention the recently reported gene-neighborhood effect and reference methods to correct for it. Specifically, the following papers should be cited: PMID: 38811841, PMID: 39030569. Along these lines, the authors should reanalyze their data using the chromosome-arm correction (AC-Chronos) processed version of DepMap data, which addresses the gene-neighborhood effect, and compare gene-buffering predictions with and without AC correction.

I also find it somewhat frustrating that the abstract immediately delves into genetic buffering and gene-pair predictions without clearly explaining what genetic buffering is. A brief explanatory sentence would provide much-needed context. Furthermore, the term "synthetic" used in the abstract is not clearly defined in this context and should be clarified.

Finally, while discussing improvements in analytical pipelines for CRISPR-Cas9 screen analysis, the following paper should be cited: PMID: 36814834.

Minor points:

- Line 57: "For instance, a cell can to cope (...)" should be corrected to "For instance, a cell can cope (...)".
- Line 62: "that" should be removed.
- Lines 65-66: The sentence here is unclear; the use of "between" is confusing and needs clarification.

Reviewer #2:

In their manuscript, the authors present BaCoN (Balanced Correlation Network), a novel method to allow more precise prediction of buffering gene pairs. The authors compare BaCoN with several existing methods with and without a priori data normalisation, using genetic perturbation and transcriptomic data from the Cancer Dependency Map (DepMap). BaCoN appears to improve performance by combining post-hoc unsupervised correction with a priori data normalisation. The topic addressed by BaCoN, namely improving the in silico prediction accuracy of buffering gene pairs, is of great importance and the method represents a potentially interesting step forward in increasing predictive power. However, I would like to raise some points that should be addressed to strengthen the claims made in the manuscript.

Major comments

1. From the data presented in Figure 2, it looks like PCC + BaCoN performs better than any other tested method in identifying potentially buffering gene pairs derived from different published reference datasets. PCC + BaCoN also seems to perform better than Cholesky whitening for most reference datasets, with the exception of DeKegel et al. validated, Thompson et al. SL and Ito et al. SL (Fig. 2). Later on, the authors show that combining BaCoN with a priori normalization methods, improves the performance of those methods, in particular in combination with Cholesky whitening (Fig. 3). However, in Fig. 4 and Fig. S5, BaCoN + Cholesky whitening was not tested anymore. What was the reason for that?
2. In line 264, the authors state: "While PCA-based whitening likely completely unlinked gene fitness effect and expression data, Cholesky whitening showed a surprisingly high number of overall and SL paralog pairs, and Cholesky whitening performance further increased when adding BaCoN (Fig. 3A; Dataset EV1)". However, in Table EV1, no predictions from Cholesky whitening without BaCoN are shown, so it is impossible to judge the benefit of adding BaCoN to Cholesky whitening. Since a major claim is that Cholesky whitening performs better with BaCoN, a side-by-side comparison of Cholesky whitening with and without BaCoN should be included in Dataset EV1.

3. In Fig. 5 and Dataset EV2, a list of 899 high-quality buffering gene pairs is presented. Here, BaCoN + Cholesky whitening re-appear as the method of choice for the prediction of buffering gene pairs. However, just like in Dataset EV1, no results are shown in Dataset EV2 or Fig. 5 for Cholesky whitening without BaCoN. Hence it is impossible to judge what benefits really come from adding BaCoN to Cholesky whitening versus using Cholesky whitening alone. I think that a side-by-side comparison of Cholesky whitening with/out BaCoN in Fig. 5 would be necessary to support the claim, that adding BaCoN to Cholesky whitening allows more sensitive prediction of buffering gene pairs. To this end, a comparison of the two candidate buffering gene pair lists (+/- BaCoN) should allow the identification of pairs that were identified specifically by the presumably more sensitive Cholesky whitening with BaCoN. Experimental validation of high-ranked candidate gene pairs - similar to what was done by DeKegel et al. (2021) for ASF1A/ASF1B and COPS7A/COPS7B - should be performed to strengthen the claim that adding BaCoN to Cholesky whitening allows the identification of buffering gene pairs that otherwise would have been missed.

Minor comments

1. I found the colors used for the bar charts in Figure 3 difficult to distinguish. Also it might help to add a label for the grey columns in the legend (e.g. without BaCoN).

Reviewer #3:

Summary:

In this manuscript, the authors proposed a method called BaCoN (Balanced Correlation Network), a post-hoc unsupervised correction method for predicting gene buffering from DepMap data. BaCoN is based on an assumption that only very few buffering partners are available for a buffered gene. The method is relatively simple yet effective for overcoming the impacts of confounding factors (e.g. lineage effect) in the prediction, such that the specificity of the prediction can be improved. Using BaCoN, the authors defined a high-confidence list of 899 buffering predictions and explored the chromosomal localization of the buffering partners.

Remarks:

The method is rationally designed and its effectiveness for data correction is proven. The authors performed a comprehensive evaluation of multiple methods using different SL lists. The application of the method is specific to the DepMap data. The top hits called with the method provides a useful resource for biomedical research. The improvement made by BaCoN seems to be incremental as compared to Cholesky whitening, a previous developed method for the same purpose, which could compromise the significance of the manuscript. I have several major concerns on i) the application to other dataset; ii) quality control of top list; iii) the interpretation of the 899 buffering predictions, where the authors claimed that "buffering genes overall are often syntenic".

Major points:

1. BaCoN was applied to the gene expression and CRISPR gene effect (processed with Chronos) data from DepMap. While the Chronos dataset is the most comprehensive gene fitness data to date, DepMap also included shRNA perturbation data (Tsherniak et al., Cell, 2017), as well as the mutation, copy number and DNA methylation profiles of CCLE cell lines. It will be interesting to apply BaCoN on these datasets, such as expression vs. shRNA or mutation vs. CRISPR. Method comparison based on the alternative datasets is preferred. The integration of multi-omics data may further improve the reliability of prediction.

2. BaCoN used an arbitrary threshold (i.e. top 100 or 1000) to determine the top hits. Which threshold is optimal? How about the hits in the rank of 1k-5k? How to control the false discovery rate in the prediction? These practical issues should be discussed in the paper.

3. The authors claimed that "buffering genes overall are often syntenic" in the abstract and the main text. This conclusion is surprising and doesn't make biological sense to me. Among the syntenic buffering pairs, are the buffered genes associated with strong correlation between expression and essentiality? Are the buffered gene and the buffering partner gene located in proximal regions of the same chromosome? If yes, these buffering predictions may be false positives because genes in proximal regions tends to be highly co-expressed among cancer cell lines. Ideally, a few of these buffering pairs should be experimentally validated to support the argument.

Minor point:

In page 13, the equation of BaCoN is provided but was not written in a proper mathematical format. Also, it is unclear to me why a correction factor k is introduced and how the value of k is chosen.

Response to reviewers

We thank all reviewers for their constructive feedback and suggestions. We have addressed all comments, and we agree these changes have improved the manuscript considerably. Below we present a point-by-point response to the reviewer's comments. Reviewer comments are colored in black; our responses are colored in blue, and changes to the manuscript are colored in red.

Reviewer #1:

The authors present BaCoN: a new computational method that outperforms existing correlation-based approaches for predicting buffering gene pairs, i.e., pairs of genes where the loss of function in one gene (due to low expression or a genomic alteration) creates a selective dependency on the second gene for cellular survival.

The authors demonstrate the effectiveness of their tool by reanalyzing publicly available data from the Cancer Dependency Map (DepMap) repository. Specifically, they use two data matrices from over 1000 cancer cell lines, capturing genome-wide expression profiles (via bulk RNA sequencing) and genetic dependencies (in terms of viability reduction upon CRISPR-Cas9 targeting). They employ a series of robust metrics, such as the number of recovered paralog-gene pairs among the top hits, to validate BaCoN's performance.

In brief, BaCoN accounts for the background covariability of each gene and post-normalizes correlation scores based on the overall correlation of a gene's expression in relation to the fitness profiles of all other genes. This approach is simple, yet elegant and effective, and it offers a valuable contribution to the identification of gene-buffering pairs from large-scale DepMap datasets. This tool will be of interest to researchers working in the field.

However, there are several points I would like to see addressed before considering this manuscript for publication.

First, the authors argue that because buffering predictions align two independent data matrices—gene perturbation fitness effects and mRNA expression—the independent, a priori normalization of these data might unlink biologically meaningful covariations. This reasoning is poorly justified and should be supported by more data-driven observations. Are there any examples of meaningful covariations that are lost when the two data matrices are processed separately but preserved through post-hoc normalization?

We thank the reviewer for raising this important conceptual point. We now clarify and further evaluate the hypothesis that independent normalization of the input data can potentially remove biologically meaningful covariation. To further illustrate this on a few examples, we added a graph for the 10 gene pairs with the strongest Chronos vs mRNA expression correlation in the original data. Those pairs should be biologically meaningful, because several of them were previously experimentally supported (Köferle *et al*, 2022; Thompson *et al*, 2021). Here, the normalized covariance (correlation) of those biologically meaningful gene pairs decreases after Cholesky whitening and completely disappears after PCA whitening (the same is NOT true for co-essentiality networks that have one instead of two independent input matrices). This is shown in new Fig. S1A-D (see below). New Fig. S1E also now exemplifies that since the global variance also decreases after data whitening, for Cholesky (NOT PCA!) whitening, z-transformed PCCs can be higher for some biologically meaningful buffering predictions but not for others (e.g. CDS1-2):

We now describe this in the results saying: “In support of the second hypothesis, we found that different a priori normalization approaches of the input data weakened or completely removed the correlation of the most strongly correlated, experimentally validated buffering paralog gene pairs including *FAM50A-FAM50B*, *RPP25L-RPP25*, *EIF1AX-EIF1AY* and *DDX3X-DDX3Y* (Köferle *et al*, 2022; Thompson *et al*, 2021), which was not the case for co-essentiality networks derived from the same matrix of Chronos scores (Fig. S1A-G, J-L).”

In conclusion, we can observe that data whitening of two independent input matrices, in contrast to a scenario where just one input matrix is normalized, CAN remove biologically relevant covariation.

We now mention the new and several existing analyses. For instance, in the results in page 7, we say: “We initially hypothesized that independent normalization of the input data prior to computing association indices could unlink biologically relevant information between the data sets. While PCA-based whitening completely unlinked gene fitness effect and expression data (Fig. 2A-E, H; Fig. S1A, B, E, K), Cholesky whitening showed a surprisingly high number of overall and SL paralog pairs, and Cholesky whitening performance further increased when adding BaCoN (Fig. 3A; Dataset EV1).” Also referring to Fig. 2:

Overall, we now clarify that a priori normalization of the input data can potentially unlink covariance and interfere with gene buffering prediction, but stress that this does not seem to be a problem for Cholesky whitening by removing some of the strong wording, for instance on page 7.

Additionally, the normalization method should be tested more comprehensively. For instance, are self-addictions (i.e., the tendency for a gene's expression to predict dependency on that gene) better detected after BaCoN correction compared to other methods that integrate mRNA expression and genetic dependency?

We now perform a more streamlined and comprehensive evaluation of the different methods. Specifically, we appreciate the reviewer's suggestion to give self-addictions, an observation we had initially termed 'self-buffering' and that we now also refer to as self-addictions (also to be in line with recent work by Pacini et al. 2024), a more central role in the evaluation of each normalization method. We now consider self-addictions as another metric that is higher if a method performs better. In the results, we describe self-addictions as follows: "Third, we considered the ability to detect a gene effect's dependency (or addiction) on its own expression level, which we refer to as self-addictions (Pacini et al, 2024)."

We added self-addictions ('self addict.') in Fig. 2E and Fig. 3E to show how BaCoN, Cholesky data whitening as well as their combination improved gene buffering predictions.

Cropped part from new Fig. 2:

Cropped part from new Fig. 3:

In new Fig. S4, which now displays this metric for the top 5000 predictions of each normalization method, we replaced the term 'self-buffering' with self-addiction to be aligned with earlier work by Pacini and colleagues.

To test the normalization methods more comprehensively, we now added another metric referred to as proximity bias of buffering predictions (see response to next concern for the investigation of the proximity bias), which likely is a driver of false positive predictions to our set of metrics (see

Fig. 2D, Fig. 3D above). To integrate the metrics with a clear biological interpretation, we build an empirical FDR, which now integrates and visualizes the performance of each normalization method or combination of methods. This empirical FDR is now described in the results in page 5 saying: “To streamline the interpretation of those six metrics, we defined gene pairs driving those metrics as true positive (TP) or false positive (FP) buffering predictions whenever plausible. We labeled predicted gene pairs in metric groups one to three as standard TP and groups one and two as strict TP. FP predictions were derived from metric groups four and five and the FDR was computed as $FDR = TP / (TP + FP)$.”. We added this empirical FDR in multiple places throughout this manuscript such as Fig. 2H, Fig. 3G (see below), Fig. 3H and Fig. S5A-E. This can now even clearer visualize the additional benefit of adding BaCoN (example Fig. 3G):

While the authors discuss factors that can obscure the interpretation of data from CRISPR-Cas9 screens (such as CN-amplification), they should also mention the recently reported gene-neighborhood effect and reference methods to correct for it. Specifically, the following papers should be cited: PMID: 38811841, PMID: 39030569. Along these lines, the authors should reanalyze their data using the chromosome-arm correction (AC-Chronos) processed version of DepMap data, which addresses the gene-neighborhood effect, and compare gene-buffering predictions with and without AC correction.

We agree with the reviewer that this is a relevant aspect of the analysis and interpretation of CRISPR-Cas9 screening data. We now cite both papers, which present a major milestone in the analysis of CRISPR-Cas9 screens. As a side note, we did not appreciate this factor sufficiently since the papers had been published just when we had prepared our initial manuscript (Lazar et al. 2024) and three days before we submitted our work (Vinceti et al. 2024). However, we feel that including an extended analysis of the methods with regard to what we refer to as ‘genomic proximity’ of buffering pairs has substantially improved our manuscript.

First, using the arm-corrected (AC) 23Q2 Chronos scores, we investigated how PCC-based buffering predictions and predictions after BaCoN, Cholesky whitening and Cholesky plus BaCoN normalization is biased towards detecting more pairs within different genomic distances. For clarification, this 23Q2 data version (with AC) was used throughout our manuscript. We found significantly higher density of pairs within a genomic proximity up to 10 Mbp for the most well-performing normalization methods only. We show this in new Fig. S3E and F (see below). We next tested how strongly Chronos score data sets with and without AC show this proximity bias in the buffering predictions. As shown in the new Fig. S3G, BaCoN but even more so Cholesky whitening add this unwanted effect:

Based on our re-analyses, we decided to define this genomic proximity bias as yet another metric – a lower score indicates better performance. In the results, we describe the proximity bias as follows: “Fifth, the effect of genomic location on CRISPR screens can be substantial (Meyers et al, 2017; Vinceti et al, 2024; Lazar et al, 2024). While the DepMap 23Q2 gene effect data has been corrected for what we refer to as proximity-driven effects and PCC-based gene buffering predictions show that this removed the increased numbers detected when using uncorrected data, several of the normalization methods we test in this work showed a strong proximity bias (Fig. S2E, F). Based on this observation, we defined gene pairs encoded on the genome within 10 Mbp of each other as potential FP predictions.”.

We added the number of proximal pairs among predictions (‘Prox. pairs’) in Fig. 2D and Fig. 3D to show how both BaCoN, Cholesky data whitening as well as their combination improved gene buffering predictions.

Cropped part from new Fig. 2:

Cropped part from new Fig. 3:

And since we can consider proximal buffering predictions as FP, we also now use this metric in our empirical FDR, for instance in Fig. 2H (see 'prox' in the y-axis label):

Finally, we removed proximal buffering predictions with a genomic distance below 10 Mbp from the list of high-confidence buffering predictions we characterize in Fig. 5 and mark all proximal predictions in the new Dataset EV1.

How BaCoN can still be useful with regard to this bias and how important arm corrections are in connection with Cholesky whitening and BaCoN is discussed on page 10 now, saying: “Genomic location and proximity of genes has been a major source of potential FP fitness phenotypes in CRISPR-Cas9 screens and those biases are controlled well in recent DepMap Chronos score releases (Meyers *et al*, 2017; Lazar *et al*, 2024; Vinceti *et al*, 2024, 2023). We found that predicting gene buffering without normalization using Chronos scores with and without arm correction (AC) did not identify a relevant fraction of pairs with genomic proximity (<1% of top 1000 predictions). However, only our top performing methods BaCoN and Cholesky whitening increased the number of unwanted proximal pairs substantially (15% - 25% of top 1000 predictions) using uncorrected

Chronos scores and benefited substantially from corrections. This suggests that additional corrections, which are routinely applied to Chronos scores, are important for improved prediction of gene buffering.”.

I also find it somewhat frustrating that the abstract immediately delves into genetic buffering and gene-pair predictions without clearly explaining what genetic buffering is. A brief explanatory sentence would provide much-needed context. Furthermore, the term "synthetic" used in the abstract is not clearly defined in this context and should be clarified.

We thank the reviewer to pointing out the lack of clarity. We extended the first sentence in the abstract to define gene buffering. It now says: “Buffering between genes, where one gene can compensate for the loss of another gene, is fundamental for robust cellular functions.”

With regard to the term “synthetic” we believe that there may be a misunderstanding. We only mentioned syntenic (genetic elements on the same chromosome) relationships between buffering predictions in the abstract. We now avoid this term in the abstract.

Finally, while discussing improvements in analytical pipelines for CRISPR-Cas9 screen analysis, the following paper should be cited: PMID: 36814834.

We agree that this work presents improvements in analytical pipelines for CRISPR screens and now cite this relevant work.

Minor points:

- Line 57: "For instance, a cell can to cope (...)" should be corrected to "For instance, a cell can cope (...)".

We have removed the word “to”.

- Line 62: "that" should be removed.

We have removed the word “that”.

- Lines 65-66: The sentence here is unclear; the use of "between" is confusing and needs clarification.

We have clarified the sentence now saying: “Similarity of expression signatures (co-expression) or genetic interaction signatures between pairs of genes have been exploited to classify genes by their function in model organisms”. We hope the sentence now conveys the intended message.

Reviewer #2:

In their manuscript, the authors present BaCoN (Balanced Correlation Network), a novel method to allow more precise prediction of buffering gene pairs. The authors compare BaCoN with several existing methods with and without a priori data normalisation, using genetic perturbation and transcriptomic data from the Cancer Dependency Map (DepMap). BaCoN appears to improve performance by combining post-hoc unsupervised correction with a priori data normalisation. The topic addressed by BaCoN, namely improving the in silico prediction accuracy of buffering gene pairs, is of great importance and the method represents a potentially interesting step forward in increasing predictive power. However, I would like to raise some points that should be addressed to strengthen the claims made in the manuscript.

Major

comments

1. From the data presented in Figure 2, it looks like PCC + BaCoN performs better than any other tested method in identifying potentially buffering gene pairs derived from different published reference datasets. PCC + BaCoN also seems to perform better than Cholesky whitening for most reference datasets, with the exception of DeKegel et al. validated, Thompson et al. SL and Ito et al. SL (Fig. 2). Later on, the authors show that combining BaCoN with a priori normalization methods, improves the performance of those methods, in particular in combination with Cholesky whitening (Fig. 3). However, in Fig. 4 and Fig. S5, BaCoN + Cholesky whitening was not tested anymore. What was the reason for that?

We agree that consistency with Fig. 2, Fig. 3 and Fig. 4 improves the manuscript and now present re-sampling results for Cholesky whitening (and other methods that performed well in Fig. 2) in Fig. 4A-C (see below) and new Fig. S6 and Fig. S7 (corresponding to previous Fig. S5; not shown below). We also added combinatorial normalization approaches to test our recommendations across different numbers of cell lines and gene spaces in Fig. 4G, H:

In the results in page 8 we now say: “We sub-sampled 100, 200, 500 screens each 10 times from the 23Q2 DepMap set of 1019 screens and tested the performance of BaCoN and the different a priori normalization methods with and without BaCoN”. And: “At 200 cell lines, all more sophisticated methods started to outperform the simple PCC, with BaCoN ComBat plus BaCoN, Cholesky whitening and Cholesky whitening plus BaCoN outperforming the other methods lines (Fig. 4A-C; Fig. S6A, C; Fig. S7A). The addition of BaCoN to Cholesky whitened data became more beneficial at 500 cell lines, particularly with regard to preventing a high number of genomically proximal, likely FP predictions (Fig. 4A-C; Fig. S6A-D). In contrast to multiple linear regression, BaCoN alone or in combination with a priori normalization was able to increase the number of predicted paralog pairs between 500 cell lines and the full data (Fig. 4A-C).”.

We initially believed that due to the strong data transformation after Cholesky whitening, ComBat normalization should be the conservative method of choice. However, the additional analyses we now added further strengthen the case for Cholesky whitening plus BaCoN.

2. In line 264, the authors state: "While PCA-based whitening likely completely unlinked gene fitness effect and expression data, Cholesky whitening showed a surprisingly high number of overall and SL paralog pairs, and Cholesky whitening performance further increased when adding BaCoN (Fig. 3A; Dataset EV1)". However, in Table EV1, no predictions from Cholesky whitening without BaCoN are shown, so it is impossible to judge the benefit of adding BaCoN to Cholesky whitening. Since a major claim is that Cholesky whitening performs better with BaCoN, a side-by-side comparison of Cholesky whitening with and without BaCoN should be included in Dataset EV1.

We thank the reviewer for pointing out how to increase the transparency of the comparison. We now added the prediction scores and the ranks of those scores as well as a newly computed empirical FDR computed after Cholesky whitening (without BaCoN) to Dataset EV1.

Apart from adding the requested information to Dataset EV1, we further investigated the benefit BaCoN brings to Cholesky whitened data. To facilitate a better comparison between all methods we have now computed an empirical false discovery rate (FDR) for the different methods. This FDR summarizes the different experimentally discovered synthetic lethal relationship as well as paralog pairs as true positives (TP) and contrasts those to probable false positive (FP) buffering predictions such as genomically proximal gene pair predictions as well as genes with a high co-expression with a strong buffering gene. This shows that BaCoN alone outperforms Cholesky alone for the highest confident predictions and that adding BaCoN to Cholesky strongly improves the performance achieved by either method alone along the entire range of, what we believe are helpful predictions (see below; new Fig. 2H and Fig. 3G). The FDR is also added to Dataset EV1.

Finally, while we now better document the improved performance when adding BaCoN to Cholesky whitening, we would like to clarify three more points about Cholesky whitening in the context of predicting gene buffering from gene perturbation effect and expression data. First, Cholesky strongly transforms the data and, as we now found strongly boosts the number of likely

FP genomically proximal gene pair predictions and adding BaCoN substantially alleviates this unwanted effect. Second, while Cholesky whitening has been used to improve co-essentiality networks derived from gene perturbation effect data, it has to our knowledge not been used and comprehensively benchmarked when contrasting gene perturbation effect and expression data to predict gene buffering. Finally, the utility of Cholesky whitening has been proven for co-essentiality studies. Since similar methods such as PCA whitening, which performed very well for co-essentiality networks, does not seem to produce biologically meaningful buffering predictions (see Fig. 2A-D; new Fig. S1A-F), our work, in addition to introducing BaCoN, also aims to provide the community with a hopefully useful benchmark of how data whitening methods can help predict gene buffering.

3. In Fig. 5 and Dataset EV2, a list of 899 high-quality buffering gene pairs is presented. Here, BaCoN + Cholesky whitening re-appear as the method of choice for the prediction of buffering gene pairs. However, just like in Dataset EV1, no results are shown in Dataset EV2 or Fig. 5 for Cholesky whitening without BaCoN. Hence it is impossible to judge what benefits really come from adding BaCoN to Cholesky whitening versus using Cholesky whitening alone. I think that a side-by-side comparison of Cholesky whitening with/out BaCoN in Fig. 5 would be necessary to support the claim, that adding BaCoN to Cholesky whitening allows more sensitive prediction of buffering gene pairs. To this end, a comparison of the two candidate buffering gene pair lists (+/- BaCoN) should allow the identification of pairs that were identified specifically by the presumably more sensitive Cholesky whitening with BaCoN. Experimental validation of high-ranked candidate gene pairs - similar to what was done by DeKegel et al. (2021) for ASF1A/ASF1B and COPS7A/COPS7B - should be performed to strengthen the claim that adding BaCoN to Cholesky whitening allows the identification of buffering gene pairs that otherwise would have been missed.

We agree that those points. We benchmarked the added value of adding BaCoN more extensively and performed validation experiments.

First, as described in response to the second comment, all scores that allow the Cholesky vs Cholesky plus BaCoN comparison are now in Dataset EV1. While new Dataset EV2 contains the new and improved list of 808 high-confidence predictions, all those pairs are also included in EV1 and can be subsetted by the filter criteria now described on page 9: "...we assembled a list of buffering gene pairs at an empirical FDR of 50% with subsequent removal of gene pairs with genomic proximity or driven by co-expression between buffering genes (both FP), as well as self-additions (TP) (see methods for details). The remaining 808 high-confidence predictions contained 227 paralog pairs, many of which had previously been experimentally identified as well as 581 gene pairs without previous description of buffering (Dataset EV2).".

Second, a more comprehensive comparison of Cholesky with and without BaCoN is now added in new Fig. 3, and also new Figures 4, S6 and S7 visualize the benefit of adding BaCoN (specifically see new Fig. 3G).

Finally, we now experimentally validated three currently unknown buffering pairs that lacked strong correlation in the original data:

We describe this observation in the results on page 7 saying: “Across all metrics, Cholesky whitening plus BaCoN substantially outperformed all other methods alone or in combination (Fig. 3A-G). To demonstrate how both Cholesky whitening and BaCoN impact the prediction of true buffering between pairs with (i) low correlation prior to normalization, (ii) no, to the best of our knowledge described synthetic lethality, and (iii) lower empirical FDR in the combinatorial normalization as compared to Cholesky alone or BaCoN alone normalization with, to the best of (Fig. 3H), we selected three gene pairs for experimental validation in a GFP competition assay (Fig. 3H, Dataset EV1). Upon CRISPR-Cas9-based perturbation of the genes individually and in combination in cultured hTERT-RPE1 cells, we observed that dual perturbation of *IDH3A* and *IDH2*, *UBE2C* and *UBE2S*, and *TRAPPC2B* and *TRAPPC2L* led to a more severe fitness effect than expected (Fig. 3I).”

We further expand the external evaluation by referring to two important and studies that have only been published after this manuscript was submitted to further support the benefit of BaCoN. On page 7, we now say: “To further test how BaCoN improves buffering predictions when added after Cholesky whitening, we compared the scores for recently described synthetic lethality. We found that the cancer patient treatment relevant relation between *PELO* and *FOCAD*, or the experimentally identified interactions between *SLC7A2* and *SLC7A1*, or *SLC7A6* and *SLC7A1* were best predicted upon combinatorial normalization (Fig. S5E; Dataset EV1) (Borck *et al*, 2025; Wolf *et al*, 2024).”, referring to new Fig. S5E:

Together, we are confident that our proposed combination Cholesky whitening plus BaCoN can pinpoint putative buffering pairs that may have been otherwise missed.

Minor comments

1. I found the colors used for the bar charts in Figure 3 difficult to distinguish. Also it might help to add a label for the grey columns in the legend (e.g. without BaCoN).

We thank the reviewer for making us aware of the unfortunate color selection. We have now changed the color panel in Fig. 3, while trying to consistently use color schemes throughout the manuscript.

Reviewer #3:

Summary:

In this manuscript, the authors proposed a method called BaCoN (Balanced Correlation Network), a post-hoc unsupervised correction method for predicting gene buffering from DepMap data. BaCoN is based on an assumption that only very few buffering partners are available for a buffered gene. The method is relatively simple yet effective for overcoming the impacts of confounding factors (e.g. lineage effect) in the prediction, such that the specificity of the prediction can be improved. Using BaCoN, the authors defined a high-confidence list of 899 buffering predictions and explored the chromosomal localization of the buffering partners.

Remarks:

The method is rationally designed and its effectiveness for data correction is proven. The authors performed a comprehensive evaluation of multiple methods using different SL lists. The application of the method is specific to the DepMap data. The top hits called with the method provides a useful resource for biomedical research. The improvement made by BaCoN seems to be incremental as compared to Cholesky whitening, a previous developed method for the same purpose, which could compromise the significance of the manuscript. I have several major concerns on i) the application to other dataset; ii) quality control of top list; iii) the interpretation of the 899 buffering predictions, where the authors claimed that "buffering genes overall are often syntenic".

We would like to thank the reviewer for the helpful remarks and have addressed them in a point-by-point response below. While we think the remarks have helped to substantially improve our manuscript, we would also like to clarify the reviewer's statement that Cholesky whitening has previously been developed for the same purpose and that the improvement is incremental. First, to the best of our knowledge, Cholesky whitening has not been developed for the same purpose, which is predicting gene buffering. It has been proven to be valuable for co-essentiality studies constructed from gene fitness effect data. As we now show and argue more precisely, data whitening of the input data when predicting gene buffering is not a trivial continuation from co-essentiality studies (new Fig S1A-F):

Therefore, we believe that our work, besides providing and benchmarking BaCoN, also provides a guide how to benchmark and apply the very powerful data whitening methods for gene buffering prediction.

Second, as we now show in new Fig. 3G (see below), adding BaCoN to Cholesky whitened data improves the performance to predict gene buffering, which is also driven by our newly added metric, the proximity bias (see new Fig 3D):

Major points:

1. BaCoN was applied to the gene expression and CRISPR gene effect (processed with Chronos) data from DepMap. While the Chronos dataset is the most comprehensive gene fitness data to date, DepMap also included shRNA perturbation data (Tsherniak et al., Cell, 2017), as well as the mutation, copy number and DNA methylation profiles of CCLE cell lines. It will be interesting to apply BaCoN on these datasets, such as expression vs. shRNA or mutation vs. CRISPR. Method comparison based on the alternative datasets is preferred. The integration of multi-omics data may further improve the reliability of prediction.

We agree that it is interesting to apply BaCoN to other data sets that connect the expression of one gene with the fitness effect of a second gene. To test the general applicability, we used simple Pearson's correlation coefficient analyses combined with BaCoN on different combinations of omics data from the DepMap portal. Again, we judged the performance based on metrics such as the number of paralog pairs among the top predictions. As expected, the Chronos buffering gene effect vs mRNA expression comparison used throughout this manuscript, performed best (new Fig. S8A). Yet, other comparisons such as the shRNA-based Demeter score vs mRNA expression detected many more paralog and experimentally identified SL pairs than expected by chance. For all comparisons that detected several expected gene pairs from one of our metrics, BaCoN generally improved the performance:

In the results on page 8, we report this by saying: “BaCoN has been designed to improve gene buffering predictions through contrasting CRISPR-Cas9 gene perturbation effect with mRNA expression and particularly excels when this is done across a large number of cell lines. We tested other omics data from the DepMap portal including shRNA perturbation screening and protein expression data to detect potential buffered and buffering genes, respectively. While CRISPR-Cas9 gene effect and mRNA expression data showed by far the best performance, adding BaCoN was beneficial when contrasting other omics data to predict gene buffering (Fig. S8A).”.

Moreover, we combined our (protein coding) gene effect (Chronos) vs (mRNA) expression with expression level of non-coding (nc)RNAs as potential buffering genetic elements. By adding ncRNA expression data to our well-explored mRNA expression data we also used the empirical FDR at the threshold corresponding to the 808 high-confidence protein coding gene pairs and predicted 52 gene-lncRNA buffering pairs. We report this starting on page 8 as:” Most of the data sets provide additional information about protein coding genes. We also tested if we could expand the list of potentially buffering genetic elements to non-coding (nc)RNAs. At a threshold determined for protein coding pairs below (see methods for details), we were able to add 52 gene-ncRNA buffering predictions, including the *MYC-MYCNO3* pair, to the list of 808 high-confidence predictions (Fig. S8B-D; Dataset EV3).” The results are shown in the new Fig. S8B-D:

Overall, we added several analysis illustrating how BaCoN performs on other omics data sets indicating that BaCoN, while being developed for a specific purpose, generalizes reasonably well.

2. BaCoN used an arbitrary threshold (i.e. top 100 or 1000) to determine the top hits. Which threshold is optimal? How about the hits in the rank of 1k-5k? How to control the false discovery rate in the prediction? These practical issues should be discussed in the paper.

We agree that selecting a biologically relevant threshold for buffering predictions is crucial. In brief, due to limitations for statistical thresholds, we now expanded our metrics and used those, where plausible, to define true positive (TP) and false positive (FP) predictions which we then used to generate an empirical FDR.

Start-of-the-art association methods provide statistical measures of confidence such as p-values. However, when we assemble a biologically reasonable empirical FDR and compare it to Bonferroni-corrected p-values derived from e.g. multiple linear regression, the statistical measures of confidence seem to be vastly inflated. For instance, likely false positive gene buffering predictions driven by genomic proximity or cell line lineage-driven co-expression patterns (see new additional and initial analyses in Fig. 1D-G; Fig. 2C, D; Fig. S3; Fig. S4) account for more than 80% of buffering predictions at a Bonferroni-corrected linear regression-derived p-value of around 1%:

Since, to the best of our knowledge, comprehensive standards for buffering predictions (similar to GO terms) are missing, we had assembled six sets of metrics. Based on those, we now established an empirical FDR that combines five of our six metrics defined for evaluating buffering predictions. In the results, we describe this on page 5 saying: “To streamline the interpretation of those six metrics, we defined gene pairs driving those metrics as true positive (TP) or false positive (FP) buffering predictions whenever plausible. We labeled predicted gene pairs in metric groups one to three as standard TP and groups one and two as stringent TP. FP predictions were derived from metric groups four and five and the FDR was computed as $FDR = TP / (TP + FP)$.”.

We used this FDR to additionally summarize the performance of all methods and combinations of methods as illustrated in new Fig. S5:

For explanation: “**Figure S5. False discovery rates for top 1000 predictions for the tested methods of predicting buffering. A. and C.** Performance of normalization methods with (A) and without (C) BaCoN using an empirical FDR which summarized five of our six performance metrics. This standard empirical FDR was used throughout this manuscript unless noted otherwise. The

FDR considering Ensembl paralogs, Ohnologs as well as experimentally validated pairs and self-additions as true positive (TP) predictions and proximal pairs and pairs driven by co-expression of the buffering gene with stronger connected buffering genes as false positive (FP) predictions. Panel A corresponds to Fig. 2H. **B.** and **D.** Performance of normalization methods with (B) and without (D) BaCoN using a stringent empirical FDR which summarized five of our six performance metrics. The stringent FDR was computed analogously, but without self-additions as TP. FP sets were kept identical for both the stringent and the non-stringent FDR. Panel C corresponds to Fig. 2G.”.

This illustration is also shown in Fig. 2H and Fig. 3G to justify our method preference:

First, this FDR provides a basis for better judging the top 100 and top 1000 buffering predictions. Second, we now used this FDR to select our set of 808 high confidence buffering pairs characterize in Fig. 5. Again, such an empirical FDR is much more stringent than statistical confidence measures. We now say: “...we assembled a list of buffering gene pairs at an empirical FDR of 50% with subsequent removal of gene pairs with genomic proximity or driven by co-expression between buffering genes (both FP), as well as self-additions (TP) (see methods for details). The remaining 808 high-confidence predictions contained 227 paralog pairs, many of which had previously been experimentally identified as well as 581 gene pairs without previous description of buffering (Dataset EV2).”.

This definition now controls the FDR of the buffering gene. We again thank the reviewer for bringing up this crucial point and believe this will substantially change how our method and results can be used and interpreted by the community.

3. The authors claimed that "buffering genes overall are often syntenic" in the abstract and the main text. This conclusion is surprising and doesn't make biological sense to me. Among the syntenic buffering pairs, are the buffered genes associated with strong correlation between expression and essentiality? Are the buffered gene and the buffering partner gene located in proximal regions of the same chromosome? If yes, these buffering predictions may be false positives because genes in proximal regions tends to be highly co-expressed among cancer cell lines. Ideally, a few of these buffering pairs should be experimentally validated to support the argument.

We agree that the finding of synteny of non-paralog buffering genes was surprising. We investigated potential issues, particularly with regard to genomic proximity and co-expression, further (Fig. 1, partially new Fig. S3, partially new Fig. S4, new Fig. S5). After identifying and removing likely FP predictions driven by genomic proximity and co-expression, all results in Fig. 5 remained stable, only non-paralogs are not syntenic anymore. While the tendency of paralogs to occur on different chromosomes became even slightly stronger, non-paralogs high-confidence buffering pairs appeared to have no tendency as shown in new Fig. 5H:

The observation of a lacking syntenic trend for non-paralogs remains stable when we control the high-confidence buffering predictions at an empirical FDR of 10%:

For transparency, we provide all predicted pairs before and after filtering for different empirical FDR, proximal pairs and co-expressed buffering genes in the new Dataset EV1 and EV2.

Finally, to establish trust in our high-confidence buffering predictions, we now experimentally validate three currently unknown buffering pairs that lacked strong correlation in the original data:

We describe this observation in the results on page 7 saying: “Across all metrics, Cholesky whitening plus BaCoN substantially outperformed all other methods alone or in combination (Fig. 3A-G). To demonstrate how both Cholesky whitening and BaCoN impact the prediction of true buffering between pairs with (i) low correlation prior to normalization, (ii) no, to the best of our knowledge described synthetic lethality, and (iii) lower empirical FDR in the combinatorial normalization as compared to Cholesky alone or BaCoN alone normalization with, to the best of (Fig. 3H), we selected three gene pairs for experimental validation in a GFP competition assay (Fig. 3H, Dataset EV1). Upon CRISPR-Cas9-based perturbation of the genes individually and in combination in cultured hTERT-RPE1 cells, we observed that dual perturbation of *IDH3A* and *IDH2*, *UBE2C* and *UBE2S*, and *TRAPPC2B* and *TRAPPC2L* led to a more severe fitness effect than expected (Fig. 3I).”

Overall, we would like to mention that likely FP buffering predictions based on genomic proximity and co-expression are now described, investigated and discussed extensively throughout the manuscript, and the value of BaCoN is clarified. For instance, in the discussion on page 10 we now say: “We show that cell lineage drives co-expression, which then leads to the identification of many false positive (FP) buffering genes, accounting for as much as 86.2% of the top 5000 predictions when only using PCC without normalization. However, taking into account lineage in multiple linear regression reduces this bias but still predicts 62% genes from this likely

FP category. BaCoN, constructed with the simple assumption that genes should be buffered by one or a few genes largely removed this bias in an unsupervised fashion.”. We continue by saying: “Genomic location and proximity of genes has been a major source of potential FP fitness phenotypes in CRISPR-Cas9 screens and those biases are controlled well in recent DepMap Chronos score releases (Meyers *et al*, 2017; Lazar *et al*, 2024; Vinceti *et al*, 2024, 2023). We found that predicting gene buffering without normalization using Chronos scores with and without arm correction (AC) did not identify a relevant fraction of pairs with genomic proximity (<1% of top 1000 predictions). However, only our top performing methods BaCoN and Cholesky whitening increased the number of unwanted proximal pairs substantially (15% - 25% of top 1000 predictions) using uncorrected Chronos scores and benefited substantially from corrections. This suggests that additional corrections, which are routinely applied to Chronos scores, are important for improved prediction of gene buffering.”.

We very much appreciate the reviewer’s comment and we believe that we have substantially improved our manuscript.

Minor point:

In page 13, the equation of BaCoN is provided but was not written in a proper mathematical format.

We thank the reviewer for helping to increase clarity of the technical presentation of our method. We believe that the reviewer has pointed to the mix of nomenclature for the sample and population correlation coefficient. We have now written the equation of BaCoN based on the sample correlation coefficient.

Also, it is unclear to me why a correction factor k is introduced and how was the value of k chosen.

Choosing k , which determines how strongly BaCoN should balance the network, has been done on a few technical metrics, which are now illustrated in new Fig. S2A-D:

We describe the selection process and the description of k in the methods, saying: “The correction factor k can range from 0 to 1, where 0 transforms the network the strongest. We chose k so BaCoN (i) detected a high number of self-additions among the top predictions and (ii) removed co-expression between buffering partners (Fig. S2A, D). At the same time, we aimed to reduce ties among BaCoN scores and favored a moderate data transformation (Fig. S2B, C).”.

17th Mar 2025

Manuscript Number: MSB-2024-12533R

Title: BaCoN (Balanced Correlation Network) improves prediction of gene buffering

Author: Maximilian Billmann

Thomas Rohde

Talip Demirtas

Sebastian Suesser

Angela Shaw

Manuel Kaulich

Dear Max,

Thank you for submitting your revised manuscript to Molecular Systems Biology. We have now received the enclosed report from two Reviewers who agreed to re-assess your work. As you will see below, both reviewers are satisfied with the revisions.

Since the original Reviewer #3 is unable to re-review the manuscript, we asked Reviewer #2 to also review the your responses to Reviewer #3's main points, Reviewer #2 thinks that these concerns have been adequately addressed. Therefore, I am pleased to inform you that we will be able to accept your manuscript pending the following amendments:

1. Please upload figures as individual figure files. The legends should be only included in the manuscript file, located below the References.
2. Please rename the code availability to "Data availability". Please also remove the "Appendix_documented_code.pdf". The computer code produced in the study should either be deposited to an appropriate public database such as GitHub, or provided as a ZIP file labelled "Computer code EV1". This ZIP file should include the data file (computer code) AND a separate plain text README file with item title and description. The ZIP file should be submitted using the file type Expanded View File in our manuscript submission system.
3. "Competing interests" should be renamed to "DISCLOSURE AND COMPETING INTERESTS STATEMENT".
4. The appendix title page should contain "Appendix for + manuscript title" and a Table of Content with the page numbers for the listed items. Nomenclature should follow the format "Appendix Figure Sx" and "Appendix Table Sx" throughout both the manuscript and the Appendix PDF.
5. Author checklist: for all positive responses, please specify the section where the information can be found (in the pink column).
6. Please download and fill our Reagents and Tools Table template (.docx), which you can find in our author guidelines: <https://www.embopress.org/page/journal/17444292/authorguide#structuredmethods>.

7. Please provide a "standfirst text" summarizing the study in one or two sentences (approximately 250 characters, including space), three to four "bullet points" highlighting the main findings and a "synopsis image" (550px width and 400-600 px height, PNG format) to highlight the paper on our homepage. Please make sure the provided image does not exceed the specified dimensions.

Here are a couple of examples:

<https://www.embopress.org/doi/10.15252/msb.20199356>

<https://www.embopress.org/doi/10.15252/msb.20209475>

<https://www.embopress.org/doi/10.15252/msb.209495>

8. Please correct the Section order should be corrected: Title page - Abstract & Keywords - Introduction - Results - Discussion - Methods - Data Availability - Acknowledgements - Disclosure and Competing Interests Statement - References - Figure Legends - Table(s) - Expanded View Figure Legends.

9. Please address the following issues related to figure legends

- Please note that information related to n is missing in the legend of figure 3I.

- Please note that the measure of center for the error bars needs to be defined in the legend of figure 4A.

When you resubmit your manuscript, please download our CHECKLIST (<https://bit.ly/EMBOPressAuthorChecklist>) and include the completed form in your submission. *Please note* that the Author Checklist will be published alongside the paper as part of the transparent process (<https://www.embopress.org/page/journal/17444292/authorguide#transparentprocess>)

Click on the link below to submit your revised paper.

Sincerely,
Jingyi

Jingyi Hou, PhD
Senior Editor
Molecular Systems Biology

Reviewer #1:

The authors have thoroughly addressed all the points I raised.
I recommend this manuscript for publication in Molecular Systems Biology.

Reviewer #2:

All my earlier concerns and comments have been addressed by the authors. In my opinion, the manuscript has improved greatly during the review process and represents an important contribution to the field. I therefore recommend the publication of the manuscript as is.

I had a look at the responses of the authors to Reviewer #3's major points 1-3. To me it looks like the authors addressed all concerns by either adding new data or discussing the issues raised by the reviewer, in the manuscript.

All editorial and formatting issues were resolved by the authors.

7th Apr 2025

Manuscript number: MSB-2024-12533RR

Title: BaCoN (Balanced Correlation Network) improves prediction of gene buffering

Dear Max,

Thank you again for sending us your revised manuscript. We are now satisfied with the modifications made and I am pleased to inform you that your paper has been accepted for publication.

Yours sincerely,
Jingyi

Jingyi Hou, PhD
Senior Editor
Molecular Systems Biology
